# Extreme melt season ice layers reduce firn permeability across Greenland

Riley Culberg [1✉], Dustin M. Schroeder [1,2] & Winnie Chu [3]

Surface meltwater runoff dominates present-day mass loss from the Greenland Ice Sheet. In Greenland's interior, porous firn can limit runoff by retaining meltwater unless perched low-permeability horizons, such as ice slabs, develop and restrict percolation. Recent observations suggest that such horizons might develop rapidly during extreme melt seasons. Here we present radar sounding evidence that an extensive near surface melt layer formed following the extreme melt season in 2012. This layer was still present in 2017 in regions up to 700 m higher in elevation and 160 km further inland than known ice slabs. We find that melt layer formation is driven by local, short-timescale thermal and hydrologic processes in addition to mean climate state. These melt layers reduce vertical percolation pathways, and, under appropriate firn temperature and surface melt conditions, encourage further ice aggregation at their horizon. Therefore, the frequency of extreme melt seasons relative to the rate at which pore space and cold content regenerates above the most recent melt layer may be a key determinant of the firn's multi-year response to surface melt.

[1] Department of Electrical Engineering, Stanford University, Stanford, CA, USA. [2] Department of Geophysics, Stanford University, Stanford, CA, USA. [3] School of Earth and Atmospheric Sciences, Georgia Institute of Technology, Atlanta, GA, USA. ✉email: culberg@stanford.edu

The extent, duration, and magnitude of annual surface melting significantly affect the mass balance of the Greenland Ice Sheet. Roughly 55% of Greenland's mass loss since 2000 is attributed to decreasing surface mass balance (SMB) driven by increasing meltwater runoff[1,2]. Runoff also impacts ice discharge as the rate and timing of meltwater delivery to the glacier bed affects basal drag and therefore ice velocity[3–6]. However, significant uncertainty remains regarding the connection of meltwater through the supraglacial, englacial, and subglacial systems and how those pathways may evolve in a warming climate.

In the high-elevation accumulation zone, surface meltwater does not flow directly to the ocean or glacier bed, but instead percolates into the porous near-surface snow or firn. Here meltwater can be stored in perennial firn aquifers[7] or refreeze locally[8,9]. Refreezing is known to form multi-meter-thick, horizontally continuous ice slabs, or ice layers (0.1–1 m thick) and ice lenses (<0.1 m thick) with limited horizontal extent[9]. The impact these features have on Greenland's ice dynamics and mass balance depends on the rate of water infiltration relative to the rate of refreezing within the firn. If deep, heterogeneous infiltration and localized refreezing dominate[10,11], then all firn pore space must be filled before runoff occurs. In this case, the lag between the onset of surface melting and the start of runoff can be on the order of decades[12].

Alternatively, if a perched impermeable horizon develops that subsequently caps otherwise permeable firn, runoff can initiate much more rapidly, enhancing surface-melt driven mass loss[13]. This scenario includes the growth of near-surface ice slabs that have significantly expanded the runoff zone in the Greenland interior[8,9]. These ice slabs are understood to be multi-annual features that initially developed below 2000 m elevation through the aggregation of pre-existing ice lenses driven by multi-year excess meltwater production[9]. In contrast, extensive near-surface ice layers linked only to extreme surface-melt in 2010 have been observed on a single transect in southwest Greenland at elevations up to 2400 m[14], suggesting that perched ice layers can form in a single extreme melt season at elevations well above the upper limit of previously detected ice slabs. Field observations also show that ice layers developed following the 2012 extreme melt season at sites as far inland as Summit Station[14–17]. However, the large-scale structural impact of refreezing on the firn is difficult to assess from point measurements alone, and such rapid formation in regions of relatively low melt suggests that the formation conditions, character, and multi-year evolution of these melt layers may differ significantly from those of previously characterized ice slabs.

In this work, we investigate the role of extreme melt seasons in melt layer formation by assessing the spatial extent and structural impact of the 2012 melt layer and the climatological conditions under which it formed, as well as its ongoing interactions with surface-melt.

## Results

**Radar sounding observations of the 2012 melt layer.** We present Greenland-wide ice-penetrating radar evidence of the extensive complex of refrozen ice layers that formed in the near-surface firn of the shallow percolation zone following the extreme melt season in 2012 (hereinafter referred to as the 2012 melt layer). This melt layer exists in almost every sector of Greenland, covering ~9000 line-km of radar sounding data collected by the Center for the Remote Sensing of Ice Sheets (CReSIS) Accumulation Radar as part of NASA's Operation IceBridge (OIB). This system, with ~0.3 m range resolution and ~15 m horizontal posting, is highly sensitive to vertical variations in firn density[9,18]. We initially identified this refrozen melt layer in radargrams collected in southwest Greenland in April 2013 that show an exceptionally bright reflector about 1 m below the surface (Fig. 1b). This reflector is absent in data collected at the same location in April 2012 prior to the extreme melt events[17] (Fig. 1a). The reflector is typically as bright or brighter than the surface return, suggesting a density contrast with the surrounding firn of at least 0.3 g cm$^{-3}$ (Supplementary Methods). This same reflector can be seen in the April 2017 radargram, now buried about 5 m deep (Fig. 1c). Based on the timing, strength, and depth of this reflector, we interpret it as a refrozen complex of ice layers, individually less than 0.3 m thick, that formed during the 2012 melt season (Supplementary Methods).

We mapped the spatial extent of the 2012 melt layer across the Greenland ice sheet by identifying similar bright reflectors within the top 15 m of the ice sheet whose power exceeds three times the standard deviation of returns from typical seasonal layering[19] (see "Methods" section). We quantified the lateral layer connectivity, the range of layer densities consistent with the radar observations, and the probability that these estimates are consistent with a layer whose density exceeds pore close-off (see "Methods" section, Supplementary Fig. 1). These metrics represent a spatial average over the approximately 20 m horizontal footprint of the radar. Lateral variations in density and layer discontinuities below 20 m likely exist but cannot be directly resolved by the Accumulation Radar.

Mapping reveals that the 2012 melt layer is pervasive in the Greenland interior at the transition between the percolation and

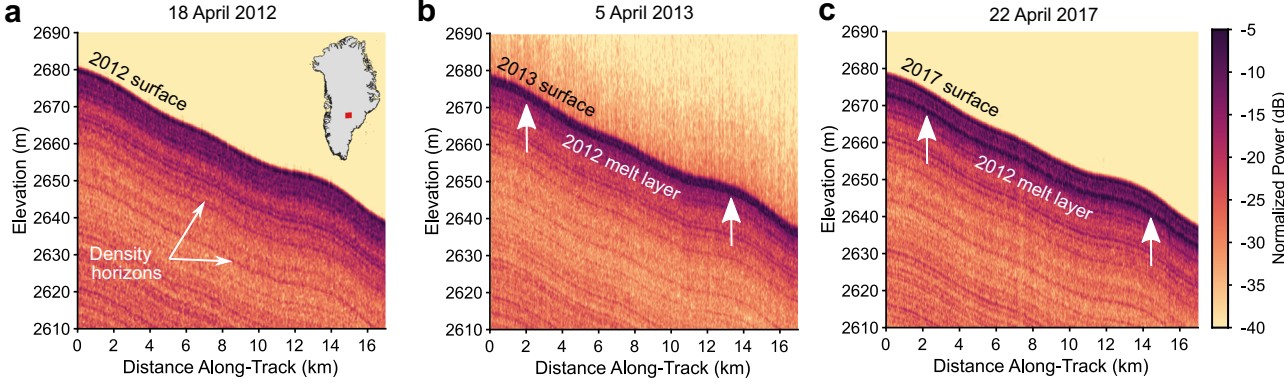

**Fig. 1 Representative radargrams from southwest Greenland showing the 2012 melt layer.** These data highlight the initial development and persistence of the 2012 melt layer in **a** April 2012, **b** April 2013, and **c** April 2017. The location of these radargrams is shown by the red line in the inset map in **a**. The color scale represents received radar power, with dark colors indicative of strong returns from areas of high density contrast within the firn.

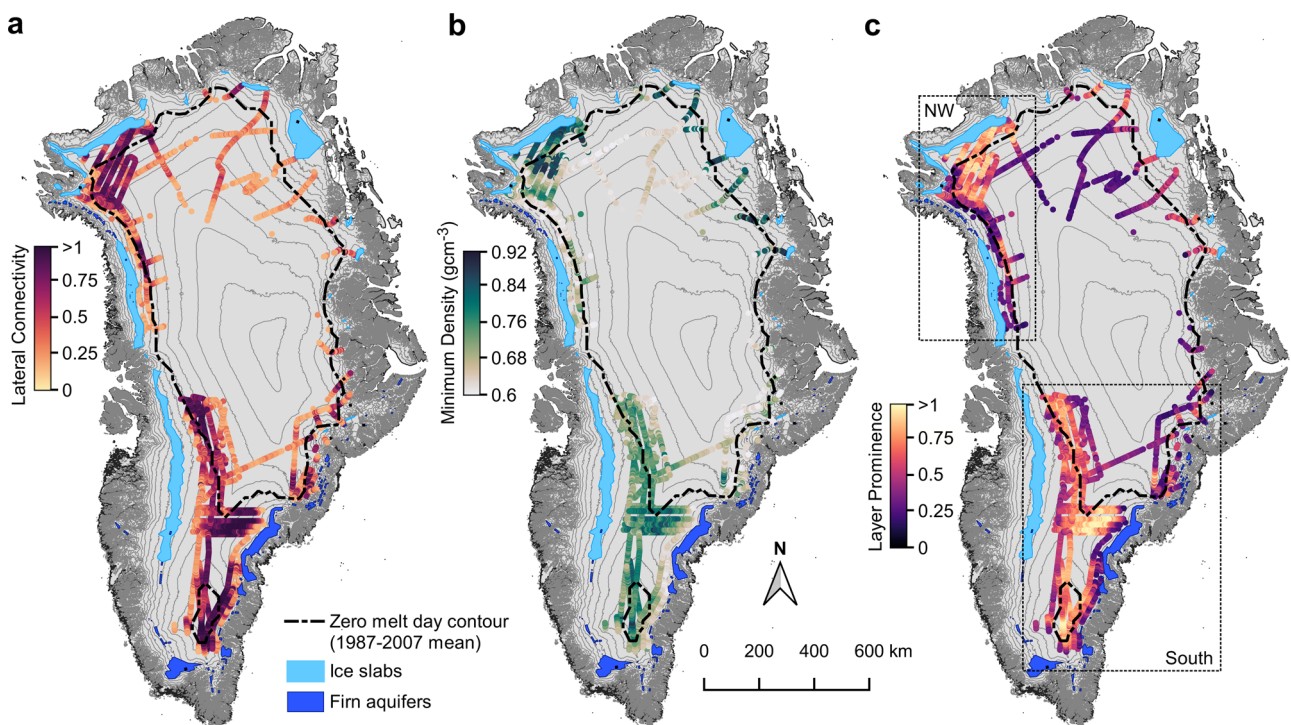

**Fig. 2 Radar-derived characterization of the 2012 melt layer. a** Horizontal connectivity of the 2012 melt layer, shown by color shading along flight transects. A score of 1 or greater indicates a perfectly connected layer, with dark colors indicating high connectivity. **b** Minimum layer density consistent with observed radar reflectivities, shown in color shading along flight transects. Dark green dots indicate high-density regions. **c** Radar-inferred layer prominence, shown in color shading along flight transects. Light yellow regions indicate high layer prominence. Dark blue polygons show firn aquifer extent[42], light blue polygons the ice slab extent[9], and the dashed black line the 1987–2007 average zero melt day contour derived from passive microwave data[64]. Surface elevation contours (200 m interval) are shown in gray. The dashed black boxes in **c** show the regions referred to as northwest and southern Greenland in the climatological analysis and Fig. 3.

dry snow zones, located at 2200–2700 m elevation in the south and 1800 m to 2200 m elevation in the northwest (Fig. 2a). The layer generally lies ~3–12 m below the April 2017 ice surface, following the regional snow accumulation gradients. Both the connectivity and density of the melt layer drop rapidly at higher elevations in the dry snow zone, where the total melt production is low even in extreme seasons (Fig. 2b). Similarly, the melt layer is generally absent at lower elevations within the deep percolation zone, except in the northwest where it extends on average 35 km down-glacier into the ice slab regions[9]. Although direct comparison of our results with firn cores is hampered by the different spatial sampling scales (see "Methods" section), both our depth and density estimates are generally consistent with the few firn core measurements that were collected in 2017 within a few kilometers of the radar lines[8,9,20] (Supplementary Fig. 2). Ice layers less than 0.02 m thick have also been observed in the field at higher elevations[17], but we do not expect this radar system to be sensitive to such thin layers (see "Methods" section). Field studies have demonstrated that total firn ice content increases significantly as elevation decreases[8,14], although the lateral layer connectivity cannot be assessed from cores. Therefore, it is possible that the melt layer exists in the deep percolation zone but is obscured by overlying infiltration ice, as our detection algorithm relies on a strong density contrast between the melt layer and surrounding firn. Altogether, our estimated extent likely represents a conservative minimum.

Next, we introduce a new combined nondimensional metric called layer prominence that describes the degree to which the firn structure and density were altered by the formation of the 2012 melt layer (Fig. 2c). This metric is calculated by taking the average of the lateral layer connectivity and the probability that

the layer density exceeds pore close-off. In other words, a solid ice layer, which is detected at the same depth in every radar trace will have a prominence of one or greater, whereas a layer prominence of zero indicates that the structure at the 2012 summer surface shows no change from the surrounding firn. Layer prominence provides a qualitative proxy for the degree to which the melt layer may present a bulk, macroscopic (>20 m horizontal scale) impediment to vertical percolation of meltwater by reducing the number of high permeability drainage pathways through the firn[21]. Further, the quantitative connection between this spatially averaged structure metric and the effective permeability of the firn would require dedicated field validation studies to measure sub-meter variability in density and microstructure, as well as the impact of preferential percolation and thermal state which is beyond of the scope of this study.

Figure 2c shows the estimated layer prominence across Greenland and reveals that the 2012 melt layer has the most significant impact on firn structure in the northwest near Camp Century, in the southern saddle, and at the southern dome. In these regions, the mean minimum density is $0.78 \pm 0.05$ g cm$^{-3}$ ($\mu \pm \sigma$), mean layer lateral connectivity is $0.82 \pm 0.17$ (where a continuous layer scores $\geq 1$), and mean layer prominence is $0.79 \pm 0.12$. The high layer prominence suggests that 2012 refreezing significantly reshaped the firn structure in these regions.

**Climate controls on melt layer formation.** To examine the climatic factors that allowed the melt layer to form in these regions, we use a three-dimensional regional climate simulation from the Modèle Atmosphérique Régional (MAR) v3.5.2[22] to investigate the spatial correlation between the radar-inferred layer prominence and climate variables. Refrozen ice structure is governed by

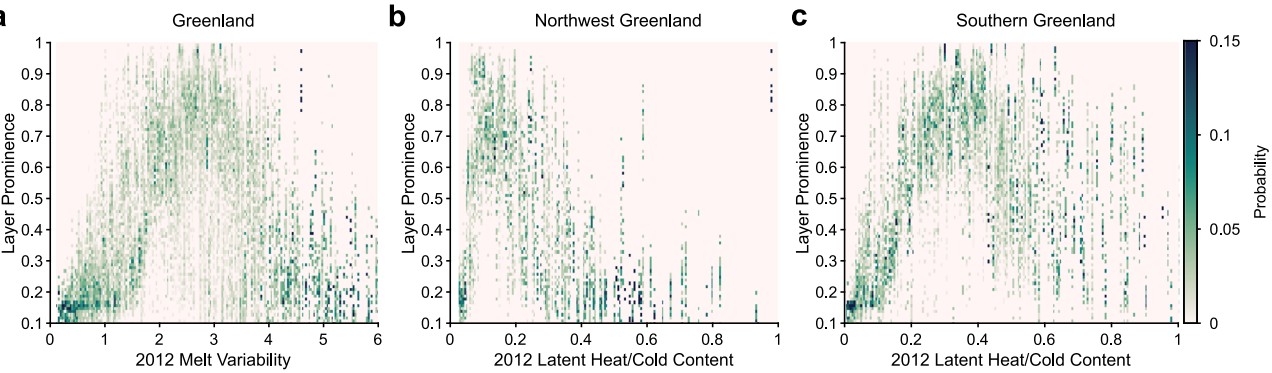

**Fig. 3 Two-dimensional histograms showing the probability of a given layer prominence occurring for a given climate condition.** Darker greens indicated higher probabilities. Greenland regions are defined by the dashed boxes in Fig. 2c. **a** Correlation with the standard deviation of the modeled daily surface-melt production from January 1, 2012 to December 1, 2012 (see "Methods" section). **b** Correlation with the ratio of latent heat in the total 2012 summer melt to firn cold content in June 2012 in Northwest Greenland. **c** Same metric for Southern Greenland.

the balance between initial firn temperatures, meltwater volume, and firn permeability[23]. Therefore, we analyze the annual accumulation, both the 2012 and mean melt to accumulation ratios, 2012 total melt anomalies relative to 1980–2011, and the ratio of latent heat in the total 2012 summer melt to the energy required to bring the firn to its melting temperature (i.e., cold content) in June 2012 (see "Methods" section) (Fig. 3 and Supplementary Fig. 3). As prior studies also suggest that the timing and duration of melt[14] as well the contrast between winter and summer temperatures[23] may play a role in ice layer formation, we also consider the standard deviation of daily melt production in 2012 as a proxy for melt variability, as well as the difference in 2011–2012 winter (December/January/February) and 2012 summer (June/July/August) temperatures (Fig. 3, Supplementary Fig. 3).

Of these metrics, we find that only the standard deviation of daily melt production during the 2012 melt season is well-correlated with the radar-inferred layer prominence at the continental scale (Fig. 3a and Supplementary Table 1). The 2012 melt season was characterized by extreme, but short-lived melt events on the order of 3–7 days[17,24] which were particularly prevalent at and above the elevations where the 2012 melt layer is present. This correlation suggests that the intra-season melt variability on timescales of days to weeks likely aided melt layer formation by producing rapid freeze-thaw cycles that promoted near-surface refreezing before deep percolation could occur[14].

However, at the regional scale, we find that the ratio of latent heat to cold content is equally or more well-correlated with layer prominence (Fig. 3b, c), although the latent heat/cold content ratio in northwest Greenland is about half that in southern Greenland for the same layer prominence. We also observe a distinct difference in the radar character of the melt layer between these sectors. In the regions of greatest layer prominence in the northwest, we generally observe one well-defined reflector, indicative of a single, continuous ice layer. In contrast, the south is characterized by a 1–2-m-thick band of bright reflectors, indicative of a conglomeration of many thin ice layers (Supplementary Fig. 4) consistent with more heterogeneous infiltration and refreezing at the meter scale[25,26]. Firn cores in the south show a similarly thick band of high-density peaks just below the depth of the 2012 end of summer surface, with low spatial and temporal coherence across the stratigraphy of cores drilled at the same sites[8,9,20], further supporting this interpretation (Supplementary Figs. 2 and 4d). Our mapping demonstrates that, rather than heterogeneous densification over the whole firn

column as we might expect from consistent deep infiltration[12], refreezing in 2012 consistently formed a laterally coherent melt layer. However, these regional differences in climate and melt layer character suggest that the mechanisms driving this kind of stratification may vary across the ice sheet.

MAR simulations of surface-melt production illuminate the mechanisms that underlie these regional differences. In the northwest, the 2012 MAR melt production was on average 8 standard deviations above 1980–2011 mean, but only 5.5 standard deviations above the mean in the south. The melt layer is most prominent in areas with a mean annual temperature around −23 °C in the northwest and −19 °C in the south. However, in these northwest regions with surface temperatures between −20 °C and −26 °C, 2012 melt production was 2–10 times higher than in regions of equal mean annual surface temperature in the south where melt layer formation was largely limited or absent. Together with the distinct character of the radar reflectors, these results suggest that the most prominent melt layer in northwest Greenland was the result of unprecedented melt production over otherwise cold firn, whereas, in the south, its formation was aided by vertical variability in firn density or microstructure[25,27] rather than sharp thermal gradients. These results also suggest that the lack of melt layer detections in the southern deep percolation zone may indicate a transition to deep, heterogeneous infiltration or meltwater penetration through temperate ice[28] that limits the formation of spatially coherent melt layers. This is consistent with the high radar-inferred density but low connectivity values we observe at the lower elevation boundary of our radar detections (Fig. 2a, b).

**Temporal and spatial evolution of the 2012 melt layer.** While we cannot directly infer a reduction of firn permeability based on the radar measurements, multi-temporal observations of the melt layer can illustrate its interactions with subsequent surface-melt. In northwest Greenland (Fig. 4a), we estimate an ~23% increase in the lateral connectivity of the melt layer between April 2013 and April 2017, while its density remained constant. As field measurements at Camp Century show no evidence for melt infiltration beyond the annual layer and find the 5 m firn temperature to be −23 °C[16,29], we assess that, in the northwest, the 2012 melt layer was largely isolated beneath subsequent accumulation.

By contrast, in some parts of southern Greenland, the melt layer likely continued to densify or accrete new ice following its formation in 2012 (Fig. 4b, c and Supplementary Methods). There

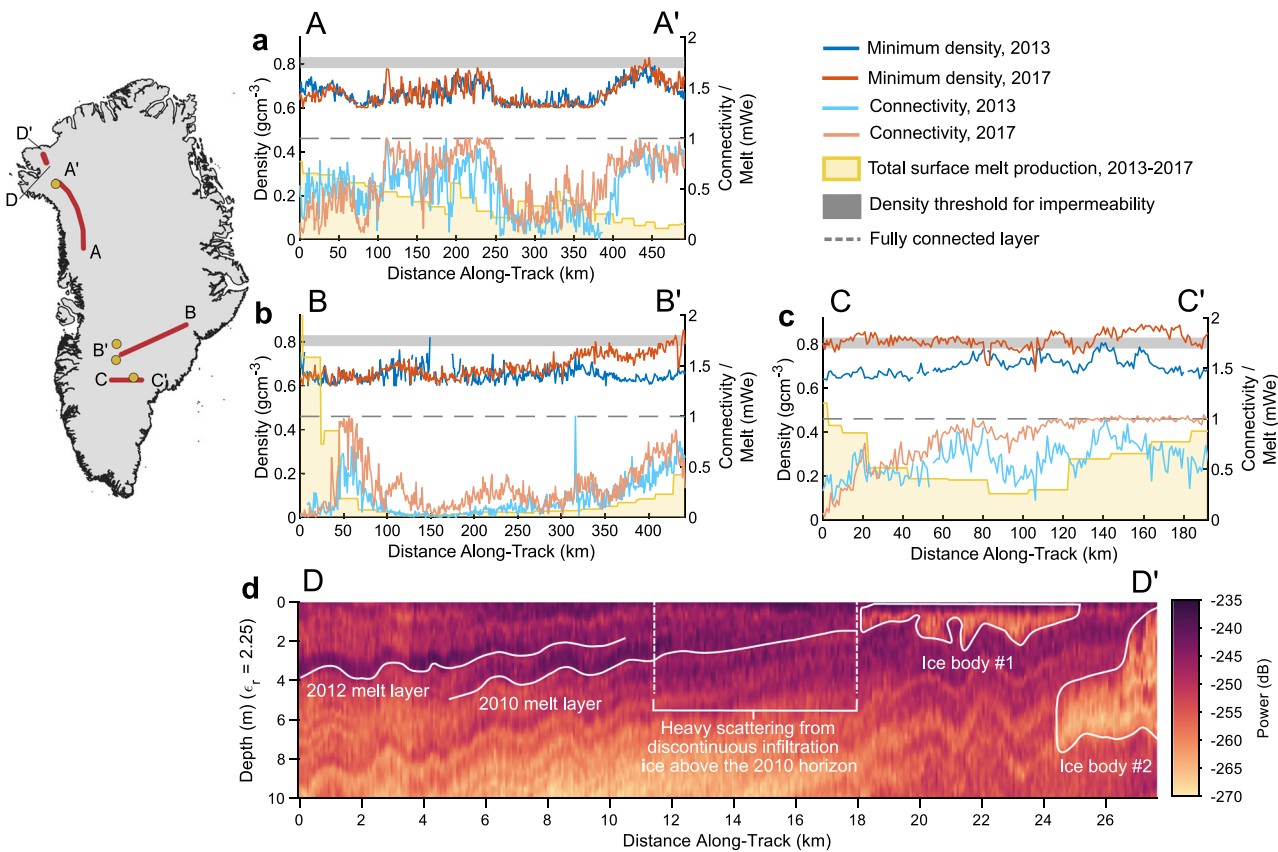

**Fig. 4 Temporal evolution of the 2012 melt layer between 2013 and 2017 along three representative flight transects where repeat flights are available.** In these panels, dark blue curves show the radar-inferred minimum layer density in 2013 and dark red curves show minimum layer density in 2017. The range of densities at which firn becomes impermeable is shown in the dark gray shading. Light blue curves show lateral layer connectivity in 2013 and light red curves show the same in 2017. The lateral connectivity score of a fully connected layer is shown in the dashed gray line. Yellow curves show the MAR-modeled total surface-melt production from April 2013 to April 2017 in meters of water equivalent. Yellow dots on the inset map show the location of firn temperature profiles from Camp Century, the NASA-SE GCNet site, and the 2016 GreenTRACS traverse and red lines show the locations of the flight transects shown in the panels. **a** Melt layer evolution in northwest Greenland. **b** Melt layer evolution from southeast to southwest Greenland. **c** Melt layer evolution in south-central Greenland. **d** Radargram showing the down-glacier convergence of the 2012 melt layer into a near-surface ice slab[8,9]. Image colors represent received radar power, with darker colors showing stronger returns consistent with high density contrasts in the firn.

we observe an increase of 0.05–0.2 g cm⁻³ in layer density along with increased lateral connectivity, both spatially well-correlated with MAR-modeled melt rates. The additional melt infiltration needed to produce this densification depends on ice layer thickness, which could range from 0.02 to 0.3 m given the sensitivity of the Accumulation Radar. Given this, the observed densification on Transect B (not accounting for compaction) would require ~1–35% of the total MAR-modeled surface-melt between April 2013 and April 2017. On Transect C, it would require ~0.5–15% of total melt. This suggests that the changes we observe would not require a consistent wetting front advance to the depth of the 2012 melt layer but could be the integrated result of transient breakthroughs from preferential percolation[10,27] over one or more years. The low spatial and temporal coherence in the stratigraphy of firn cores from this region[8,9,20] is consistent with this mechanism (Supplementary Fig. 2).

While the field observations of preferential percolation depths are limited and from transects much lower in the percolation zone[10,30–32], breakthrough percolation has been observed in firn as cold as −10 °C and around 2 m ahead of the wetting front[10]. Given these observations, firn temperature measurements from the early 2000s at the NASA-SE GCNet site[33,34] and the 2016 GreenTRACS traverse[15,35] suggest that meltwater percolation to

depths of up to 2 m during the melt season could be possible on these transects. Altogether, in the central and eastern segments of each transect, the observed densification (0.05–0.1 g cm⁻³) is most consistent with additional refreezing within thin (<0.1 m) layers sometime during the 2013 melt season. The greater densification (0.1–0.2 g cm⁻³) in the southwest may represent infiltration and additional refreezing over multiple years within thicker (up to 0.3 m) layers, consistent with the higher melt production, warmer firn temperatures, and shallower depth of the melt layer in this region. Of particular note, 2016 modeled melt production in the southwest was 2–3 standard deviations above the 1980–2011 mean with the 2012 melt layer buried only ~3.5 meters deep at the time.

These observations of continued densification or ice accretion imply that, in some parts of southern Greenland, the 2012 melt layer was of sufficiently low permeability to concentrate new refreezing at or above its horizon, leading to the continued strengthening of this aquitard even during the subsequent typical melt seasons. Similarly, in south-central Greenland, we find evidence that the 2012 melt layer developed atop an existing melt layer that initially formed following extreme melt in 2010 (Supplementary Fig. 5). This is consistent with the 2010 melt layer previously identified southeast of the Expédition

Glaciologique Internationale de Groenland (EGIG) line[14]. The multi-year persistency and continued growth of these refrozen features highlight a feedback mechanism where, under appropriate firn temperature and surface-melt conditions, existing melt layers can promote the ongoing development of low-permeability horizons by reducing vertical percolation pathways and concentrating refreezing near the surface, even if the original melt layer is not itself fully impermeable. This is analogous to the ice lens aggregation mechanism proposed for the growth of ice slabs[9], but in these interior regions, an extreme melt season can be the key catalyst that initiates this aggregation, rather than a multi-year excess melt. This implies that the frequency of extreme melt seasons relative to the rate at which the long-term melt to accumulation ratio allows new pore space and cold content to regenerate above the most recent melt layer[36] can alter the water storage capacity of near-surface firn and its response to ongoing surface melting in the ice sheet interior. This timescale is critical because Greenland has already experienced five recording-breaking melt seasons since 2000[17,37–40], with the 2019 season second only to 2012 in total melt extent[40]. Therefore, while the 2012 melt layer will eventually be isolated below new accumulation, this trend suggests that new extreme melt season ice layers will continue to form, making these processes of continued relevance.

Where the melt layer can remain in contact with active hydrology, the presence of a low-permeability barrier in the Greenland interior impacts the near-surface hydrology and other downstream refreezing features in the percolation zone. Using observations from the Alfred Wegener Institute's 2016 ultra-wideband radar survey, we find evidence for down-glacier convergence of the 2012 and likely 2010 melt layers into a near-surface ice body roughly 1–2-m thick in northwest Greenland where ice slabs were previously identified[9] (Fig. 4d). The spatial connection between these two features indicates that interior melt layers likely serve as a sufficient (but not necessary) foundation for the development of near-surface ice slabs. A similar interconnection has been documented on the Devon Ice Cap, where ice slabs developed atop refrozen ice layers formed by extreme melt in 2005 and 2010[41].

In southeast Greenland, there is a similar interaction between the 2012 melt layer and downstream firn aquifers at Helheim Glacier[7]. Here, Miège et al.[42] presented ground-penetrating radar evidence of water migrating laterally along an ice layer perched above the main aquifer body. We find that the depth of this perched water table is consistent with the modeled depth of the 2012 melt layer at the time that data was collected (see "Methods" section and Supplementary Table 2), suggesting this melt layer may have aided the development of perched aquifers by providing a low-permeability foundation in these high accumulation regions.

## Discussion

These down-glacier connections to ice slabs and firn aquifers highlight how the multi-year impact of extreme melt season ice layers depends on the local SMB regime, with outcomes ranging from vertical isolation to the development of near-surface ice slabs. Similarly, the different degrees of melt layer development that we observe across regions with similar mean annual temperatures and melt to accumulation ratios implies that melt layer formation may be strongly governed by local, short-timescale thermal and hydrologic processes in addition to the mean climate state. At the same time, our findings reveal that extreme melt layers form in nearly all regions of the Greenland Ice Sheet, with interactions spanning facies from the dry snow zone to the wet snow zone.

Given that the frequency, intensity, and/or duration of heatwaves are very likely to increase by the end of the twenty-first century[43], extreme melt layers are poised to play an increasingly important role in ice sheet hydrology and mass balance. In vulnerable regions, the character and frequency of extreme melt seasons in Greenland will be a significant factor determining firn storage capacity, since the more frequently these layers form, the more capacity may be reduced. Once formed, where these layers remain in contact with the surface hydrology, they concentrate new refreezing at or above their horizon, amplifying the contribution of even typical subsequent melt seasons to the development of perched, low-permeability horizons. As a result, these layers can aid the inland expansion of both ice slabs and firn aquifers, ultimately promoting lateral meltwater flow over local storage. Our results demonstrate that extreme melt seasons like 2012 do more than merely amplify typical SMB processes within that season. Instead, they produce persistent structural changes that can interact with the longer-term atmospheric forcing to reshape the near-surface hydrology of the accumulation zone, impacting the ice sheet's response to subsequent surface melting for years to come. Therefore, extreme melt season ice layers stand to be a key determinant of Greenland's future mass balance, impacting both surface mass balance and hydrological controls on ice dynamics.

## Methods

**Radar System and Data**. The radar data used in this study were collected by the CReSIS Accumulation Radar, a chirped pulse airborne radar that operates at a center frequency of 750 MHz with 300 MHz bandwidth[44]. We use the L1B radargrams available from ftp://data.cresis.ku.edu/data/accum/. We analyze ~22,000 line-km of data from the 2017 season, coincident flight lines from the 2013 season, and selected flights from the 2012 season for comparison and calibration. Data were collected between late March and early May in all seasons. We exclude flight segments from regions previously characterized as firn aquifers or ice slabs, as well as data below the long-term equilibrium line as defined by MARv3.5.2 [22].

**Data calibration**. To compare observed radar power with modeled radar reflectivity, the radar data must be absolutely calibrated to account for englacial attenuation, rough interface scattering, flight altitude, and variable system gain[45]. We neglect englacial attenuation and rough interface scattering as these factors likely introduce small (~1 dB) variations in the near-surface, and because any correction would increase the apparent density of the melt layer (see Supplementary Methods for further discussion). We correct for geometric spreading using a range squared dependence and radar-derived flight clearances. To account for variable system gain, which is assumed constant within each flight track, we cross-level the flight tracks within each season[46]. We then simulate the expected subsurface reflectivity at the B18, B26, and B29 firn core sites from the North Greenland Traverse using high-resolution density measurements[47–49] input to a 1D multilayered dielectric model of radar scattering[19]. Finally, we solve for the system offset value that minimizes the mean square error between model and radar observation for every measurement within 1 km of each firn core site and take the mean of those solutions as the true system offset (Supplementary Methods).

**Layer detection**. We apply a threshold detection algorithm[19] to the first 50 samples (approximately top 15 m) of the data to detect potential ice layers. We select all radar returns between the surface and the depth where the 20-point running mean of the radar trace is <30 dB above the noise floor or the last sample before the surface multiple, whichever occurs first. We detrend the log power data using a 5th order polynomial fit to the mean of the upper and lower envelope of the trace and take the standard deviation of the deepest two-thirds of the data to be indicative of typical seasonal variability. Any layer peak which exceeds three times this seasonal variability threshold is considered a melt layer detection. This threshold has been shown in simulation to detect 78% of ice layers between 0.02 and 0.3 m thick, a limit set by the physics of the imaging problem rather than an algorithm or threshold selection[19]. We exclude all measurements where aircraft roll exceeds 0.05 radians, as they are radiometrically unreliable due to power loss from the antenna beam pattern.

This simulation method has been shown to effectively reproduce Accumulation Radar data from North Greenland Traverse core sites where sufficiently high-resolution density data has been collected[18]. For layer detection algorithm validation, we use a statistical background density model based on those cores to model realistic seasonal variations in density, set the mean density profile based on

low-resolution cores from our regions of interest, and use a density for refrozen ice determined by in-situ measurement[8,19].

In general, we expect that this layer detection method produces a conservative estimate as it relies on a measure of ice layer brightness which may be suppressed by roughness, discontinuities, variable density, or receiver noise in real radar data. Additionally, the density inversion described below allows us to better constrain the physical conditions rather than assume that every detection indicates a layer whose density is that of solid ice.

**Layer connectivity**. We assess the lateral connectivity of the melt layer using a $3 \times 3$ moving kernel centered on each melt layer peak detected from the radar data. Every adjacent detection in the far right or left columns of this kernel receives one point and this score is normalized by the expected score for a fully laterally connected layer (Supplementary Fig. 1). The maximum possible connectivity score is 5/3, with any score equal to or >1 indicating that a detected layer peak is laterally continuous across the three measurements.

*Reflectivity Inversion*. To estimate the density of the melt layer, we invert the observed radar reflectivities by comparison with radar simulations. We assume that the dominant reflector in each range bin is a single dense layer of thickness between 0.02 and 0.3 m which is flat and continuous over the system's first Fresnel zone (~20 m). We use a statistical density model[19] with a variability envelope of $0.04$ g cm$^{-3}$ to simulate the background firn density profile. We then simulate the expected mean layer reflectivity for a given layer density, thickness, and background firn density using a 1D multilayered dielectric model of radar scattering[19]. We run simulations for a parameter space consisting of layer densities from 0.6 to 0.92 g cm$^{-3}$ in 0.01 g cm$^{-3}$ steps, layer thicknesses from 0.01 m to 0.3 m in 0.005 m steps, and background densities from 0.35 to 0.55 g cm$^{-3}$ in 0.05 g cm$^{-3}$ steps.

For each melt layer peak detected by our threshold algorithm, we compare the observed, calibrated radar reflectivity with the simulated reflectivity at the appropriate background density to constrain the possible layer thickness and densities consistent with observation. We derive the expected background firn density from a biexponential fit to the lower bound of the April 2012 firn density as modeled by the Institute for Marine and Atmospheric Research Utrecht (IMAU) Firn Densification Model[50].

We find that we cannot meaningfully constrain layer thickness, as the range of consistent thicknesses typically spans most of the parameter space. We bound the density in two ways. One is the absolute minimum density that is consistent with the observed reflectivity. This is the maximally conservative metric for the layer density. Second, we calculate the probability that the observed radar reflectivity is consistent with a melt layer whose density exceeds 0.81 g cm$^{-3}$, a mean density at which firn becomes impermeable[51]. We do this by taking the ratio of the number of consistent solutions which require a density in excess of 0.81 g cm$^{-3}$ to the total number of consistent solutions within the parameter space.

Similar to the layer detection algorithm, inversion by empirical comparison would require a robust collection of spatially and temporally coincident field observations spanning most reasonable configurations of layer thickness, spacing, density, and background density. Inversion by comparison to forward modeling allows for a more robust exploration of the parameter space. However, given the assumptions necessary to this process, we report the most conservative metric of the minimum density consistent with our observations.

**Post-processing**. To semi-automatically extract only the 2012 melt layer from the processed data, we calculate the approximate depth of the 2012 summer surface with the Herron and Langway model[52], extract all detections within ±4 samples (~1.2 m) of that depth and manually confirm that this region included the distinct melt layer. Frequently, the 2012 melt layer consists of a 1–2 m thick stack of ice layers, so we collapse the layer in the vertical by taking the metric maximum within each trace. We then average over 1 km horizontal bins. If a 1 km bin has <10 valid samples in the horizontal (due to aircraft roll or lack of detections), we set that bin to NaN.

We finally calculate the layer prominence metric according to Eq. 1, where $L_p$ is the layer prominence, $L_c$ is the average layer lateral connectivity (value from 0 to 5/3) in the 1 km bin, and $L_I$ is the average probability that the layer density exceeds 0.81 g cm$^{-3}$ (value from 0 to 1) in the 1 km bin.

$$L_p = 0.5(L_c + L_I) \tag{1}$$

**Comparison with in-situ observations**. True validation of radar-inferred parameters against in-situ observations would require many firn cores that are spatially and temporally coincident with the radar transects, ideally with multiple measurements within the radar footprint. Existing observations collected for other purposes largely do not meet these requirements. However, given the significant extent of the 2012 melt layer, we can at least qualitatively compare our results to observations within a few kilometers of the radar flight lines. The radar data is primarily responsive to sharp vertical variations in dielectric constant and therefore density but represents a spatial average over the horizontal footprint (about 20 m for this system). By contrast, firn core density measurements are a vertical average over the sampling depth but representative of only a single point in space. As a result, these measurements are not entirely equivalent, but the qualitative comparison does provide a sense for whether our radar-derived parameters are

consistent with the physical state of the ice sheet at a small scale. Supplementary Fig. 2 shows these comparisons, largely from the southwest and south-central Greenland.

**Climate variable correlations**. We quantify the capacity of the upper twenty meters of the firn pack to refreeze the surface meltwater produced in summer 2012 by comparing the latent heat of total melt production to the cold content in the top 20 m of firn on June 1, 2012:

$$R = \frac{\mathscr{L} M \rho_w}{20 c \rho_f T_f} \tag{2}$$

Here $\mathscr{L}$ is the latent heat of fusion, $M$ is the total melt in meters of water equivalent, $\rho_w$ is the density of water, $c$ is the heat capacity of ice, $\rho_f$ is the volume-weighted mean firn density, and $T_f$ is the absolute value of the mass-weighted mean firn temperature in degrees Celsius[35]. $M$, $\rho_f$, and $T_f$ are derived from the MARv3.5.2 3D model outputs. The MAR simulations were forced at the boundaries by NCEP-NCAR reanalysis data, and the output is provided at 20 km spatial resolution. The depth resolution varies from 10 cm in the upper 2 m to 5 m in the deeper portion of the firn. We quantify the variability in 2012 summer melt rates as the standard deviation of the daily melt production time series in each grid cell from the same MAR model run. We quantify the correlation between the radar-inferred layer prominence and each climate variable using the distance correlation coefficient[53] since the relationships are non-linear, non-monotonic, and not easily parameterized (Supplementary Table 1).

**Ultra-wideband radar**. This radargram was collected by the AWI Ultra-wideband Radar Depth Sounder which operates at a center frequency of 335 MHz over a bandwidth of 370 MHz for a theoretical range resolution of approximately 0.27 m in dry firn. We use the L1b product provided by CReSIS. To improve the visual interpretability of Fig. 4d, we up-sample in range by a factor of 10 using cubic spline interpolation, retrack the surface, and then flatten the radargram using nearest-neighbor interpolation. We then incoherently average an additional 5 times along-track. Ice slab thickness is calculated assuming an index of refraction of 1.78.

**Depth to perched water table**. We estimate the April 2014 depth of the 2012 melt layer near Helheim Glacier using the University of Washington Community Firn Model[54]. Since layer burial depth only depends on accumulation and melt rates after layer formation, we run a 50-year spin-up using 1980–2011 mean annual accumulation rates and surface temperatures as simulated by MAR at 66.1796° N, 39.0225° W. We then run the model from 1 Sep 2012 to 1 April 2014, forced by daily snowfall rates, melt rates, and surface temperature values simulated by MARv3.5.2. We estimate surface density from a spatial parameterization[55] and use a bucket refreezing scheme[56]. We implement five model runs, each with a different densification scheme: Herron and Langway dynamic densification[52], CROCUS model[56], Arthern 2010 transient[57], Barnola[58], and Kuipers-Munneke[50]. Across these five models runs, we estimate the mean depth of the 2012 melt layer to be 7.7 ± 0.4 m ($\mu \pm \sigma$) We measure the distance from the surface return to the bottom of the perched water table return in Figure 12 from Miège et al.[42] in 12 locations and estimate the mean depth of the water table in the radar data to be 7 ± 0.3 m.

## Data availability

All radar-sounding data used in this study are available from the CReSIS public FTP page: ftp://data.cresis.ku.edu/data/accum/. High-resolution firn core density profiles used for radiometric calibration are available from PANGAEA[47–49]: https://www.pangaea.de/. MARv3.5.2 climate model outputs and metadata[59] are available from the NSF Arctica Data Center at https://arcticdata.io/catalog/view/doi%3A10.18739%2FA2H12V80V. Depth-corrected GCNet temperature profiles[60] are available from the NSF Arctic Data Center at https://arcticdata.io/catalog/view/doi%3A10.18739%2FA2833N00P. GreenTRACS temperatures profiles[61] are available from the NSF Arctic Data Center at https://arcticdata.io/catalog/view/doi%3A10.18739%2FA2ZC7RV70. IMAU FDM model outputs are available by request from imau@science.uu.nl due to the large data volume The context data used in Fig. 2 are available as follows: surface elevation—Present Day Greenland compilation (http://websrv.cs.umt.edu/isis/index.php/Present_Day_Greenland), administrative boundaries—Global Administrative Areas 2015 (v2.8) (https://geodata.lib.berkeley.edu/catalog/stanford-sd368wz2435), ice slab regions[62] (https://doi.org/10.6084/m9.figshare.8309777), firn aquifer regions[63] (https://doi.org/10.18739/A2985M), and zero melt day contour—Greenland Ice Sheet melt characteristics derived from passive microwave data[64] (https://doi.org/10.5067/NON9395MQ9TK). The results of this study[65] (geolocated layer connectivity, minimum consistent density, probability of impermeability, and layer prominence) are available through the NSF Arctic Data Center at https://doi.org/10.18739/A2736M33W. Source data are provided with this paper.

## Code availability

The MATLAB scripts used for the radar analysis[66] described in this study are available through Github (https://github.com/rtculberg/2012GreenlandMeltLayer) and archived at Zenodo (https://zenodo.org/record/4552834).

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

## Acknowledgements

R.C. was supported by a National Defense Science and Engineering Graduate Fellowship. R.C., D.M.S., and W.C. were supported in part by NASA Grant NNX16AJ95G and NSF Grant 1745137. We acknowledge the use of data from CReSIS generated with support from the University of Kansas, NASA Operation IceBridge grant NNX16AH54G, NSF grants ACI-1443054, OPP-1739003, and IIS-1838230, Lilly Endowment Incorporated, and Indiana METACyt Initiative. We thank Peter Kuipers-Munneke for providing the IMAU FDM model output and Matthew Siegfried for comments on the manuscript draft.

## Author contributions

R.C. and D.M.S. conceived the study. R.C. conducted the radar data processing and analysis, climate analysis, and firn modeling. D.M.S. contributed to the development of radar processing methods. R.C., D.M.S., and W.C. all contributed to the scientific interpretation of the results and the writing of the manuscript.

## Competing interests

The authors declare no competing interests.
