## [Peer Review File · Nature Communications]

REVIEWER COMMENTS

Reviewer #1 (Remarks to the Author):

Culberg et al. present an interesting study on a very relevant topic. Climatic change influences the firn layer of the Greenland ice sheet and leads to the formation of ice slabs. The trigger of ice slab formation, however, remains poorly understood. The manuscript addresses this important topic and attributes major significance to extreme melt events, in particular the summer of 2012.

General remarks

The conclusions of the manuscript might be valid; however, they seem poorly supported by the data. I agree that Operation IceBridge (OIB) radar measurements show widespread evidence of an ice layer that formed in the firn during the summer 2012. However, I do not fully understand how the study reaches its further conclusions, such as quantifying permeability of the firn, accretion of ice around the 2012 melt layer and identification of “fundamentally different [...] mechanisms” at play (lines 130-131). These findings might be correct, but they seem poorly supported by the data and they lack critical evaluation. Furthermore, I do not understand why the study relies almost exclusively on remote sensing and modelling: none of the findings seem critically assessed in the context of field measurements. The changing firn of the Greenland ice sheet is currently a very active research field (e.g., *Humphrey et al., 2012; Forster et al., 2014; Miège et al., 2016; Graeter et al., 2018; MacFerrin et al., 2019*) and a broad variety of field measurements has become available over the last decade (e.g., *Harper et al., 2011; Machguth et al., 2016; Montgomery et al., 2018; Vandecrux et al., 2019; Karlsson et al., 2019*).

For example, at Camp Century the authors identify a strong reflection from the 2012 melt horizon. Furthermore, it is stated that firn permeability in the area is reduced by 56 ± 14 %. By coincidence, I was present during the drilling of two firn cores (73 and 62 m deep) at Camp Century, in the summer of 2017 (only a few months after the OIB overflights used in the manuscript). Indeed, the 2012 melt event was clearly recognizable in the core and appears as two ice lenses, each of them about 5 cm thick and about 8 cm apart (Fig. 1, data available upon request at <http://www.campcenturyclimate.dk>). These were the most massive ice lenses observed in the cores (see, e.g., *Karlsson et al., 2019*) and Culberg et al. identify them successfully from OIB data. However, given that there are only two very thin ice lenses above the 2012 horizon (nearly no melt since 2012; *Karlsson et al., 2019*) and that firn temperature at the depth of the 2012 lenses is roughly -23 °C (*Colgan et al., 2018*; also see <http://www.campcenturyclimate.dk/ccd/data.html>; data available at <http://promice.org/>), I doubt that the connectivity of the ice lenses changed after 2012.

Lines 36-38 read “Our results demonstrate that a single extreme melt season can reshape the near-surface hydrology of Greenland’s interior, altering the ice sheet’s *long-term response* to surface melting.” (lines 36-38, emphasis added). In its current form, I do not think the study provides sufficient evidence to support this statement. My criticism (i) touches upon the use or quantification of “permeability” (see detailed comments) and (ii) the relevance of isolated (in the vertical domain) ice lenses in a nearly melt-free environment. In the current climate, these ice lenses get buried by accumulating snow, their potential to block meltwater percolation seems to be of theoretical nature

(as illustrated on the example of Camp Century where five years after the 2012 melt event the lenses are at 5 m depth and there is no evidence of any other major melt event after 2012).

Fig. 1: The two ice lenses that likely formed in 2012, observed in the 73 m ice core at the former location of Camp Century (drilled July 25 2017 at 77.1826 °N; 61.1125 °W, 1886 m a.s.l.). The two lenses are about 8 cm apart, with the upper ice lens at 5.0 meters below the surface (5.3 m in a second core, 62 m deep and drilled at a distance of ~1.5 km).

Detailed Comments

Lines 78 – 79: “Density contrast of 0.3 g cm^{-3} ”. I do not know whether this value is also valid for the Camp Century location, but the measurements at Camp Century roughly confirm such a contrast: firn density in the cores was measured at 0.1 m vertical resolution, the three 0.1 m section that cover the 2012 ice lenses in the 73 m core have densities of 0.709 , 0.508 and 0.64 g cm^{-3} . Density of the surrounding firn is roughly 0.45 g cm^{-3} .

Lines 87 – 90: Unfortunately, I do not understand how “permeability” is calculated (also the description in the methods section is unclear to me). Please also consider that there is rich evidence that ice lenses are basically never “impermeable” (e.g. Hughes and Seligman, 1939, pages 635-640; Samimi et al., 2020). Any extended ice lens seems to have sufficient discontinuity (cf. Brown et al., 2011) to allow for percolation of water. Machguth et al. (2016) even show evidence of minor amount of meltwater percolating to >5 m depth in an ice slab environment.

Lines 111 – 113: “This substantial reduction in permeability suggests that the melt layer has the potential to impact near-surface hydrologic processes over a wide swath of the ice sheet where the layer exists.” See my major comments and the comment above. In my opinion, such a statement needs more evidence or at least an in-depth discussion.

Lines 132-141: I agree to these findings. However, I would not label them “fundamentally different mechanisms” (line 131). I expect formation of a less distinct ice layer where meltwater percolates through firn that has seen more regular percolation in the past.

Lines 152-156: See my major remarks. I understand that the Camp Century example cannot simply be considered representative for all of the ice layer formation in the north-west (although the location is

highlighted in the manuscript). Nevertheless, more information is needed to explain how this would work.

Lines 156-161: It is stated earlier in the manuscript that in 2017 the 2012 ice layer was 3 to 12 m below the surface, depending on accumulation rates. I assume that 12 m is more valid for the south where accumulation is higher? Please explain whether the suggested growth in ice thickness at substantial depth (up to 6 m on average) is possible. It might be possible, but needs to be discussed in the context of meltwater percolation depth under given amount and frequency of melt after 2012 as well as observed firn temperatures.

Lines 165-171: See my major comments and the comment above: This important statement needs more evidence.

Lines 171-173: I basically agree to this statement. However, mentioning the extraordinary melt season of 2019 might be confusing in the context of this study. All firn changes described refer to the melt seasons of 2016 or earlier.

Lines 195-196: I agree that bursts of melt play an important role in ice layer formation. However, the study also states that firn cold-content is relevant (here defined as the total cold content over the top 20 m, lines 388-390). The cold content of the top 20 m is strongly controlled by climate.

Lines 196-197: This statement is not fully clear to me.

Figure 3: The subplots lack labelling a), b) and c). How are the different regions of Greenland defined?

Methods, data calibration and layer detection: I might be wrong, but would more recent *in situ* firn measurements be helpful to calibrate and validate interpretation of the OIB data?

Lines 346, 357, 367: Please explain why data interpretation seems to be validated solely against idealized model output. In the end, the OIB data are used to make statements on real-world processes on the Greenland ice sheet. And there is a large amount of *in situ* firn measurements available.

Extended Data Figure 2: In general, the handling of geographical locations is confusing. Looking at this figure, I am surprised that a radar transects slightly north of 70 °N is labelled “southwest Greenland”. But I agree with the naming if it were somewhere stated or shown how the authors subdivided Greenland for the purpose of this study.

Extended Data Figure 3: Similar to above, for the most part this transect seems to be located in the east, not in the west.

References

- Brown, J.; Harper, J.; Pfeffer, W.; Humphrey, N. & Bradford, J., High Resolution Study of Layering within the Percolation and Soaked Facies of the Greenland Ice sheet, *Ann. Glaciol.*, 2011, 52, 35-41
- Colgan, W.; Pedersen, A.; Binder, D.; Machguth, H.; Abermann, J. & Jayred, M., Initial field activities of the Camp Century Climate Monitoring Programme in Greenland, *Geological Survey of Denmark and Greenland Bulletin*, 2018, 41, 75-78
- Forster, R. R.; Box, J. E.; van den Broeke, M. R.; Miège, C.; Burgess, E. W.; van Angelen, J. H.; Lenaerts, J. T. M.; Koenig, L. S.; Paden, J.; Lewis, C.; Gogineni, S. P.; Leuschen, C. & McConnell, J. R., Extensive liquid meltwater storage in firn within the Greenland ice sheet, *Nat. Geosci.*, 2014, 7, 95-98

Graeter, K. A.; Osterberg, E. C.; Ferris, D. G.; Hawley, R. L.; Marshall, H. P.; Lewis, G.; Meehan, T.; McCarthy, F.; Overly, T. & Birkel, S. D. Ice Core Records of West Greenland Melt and Climate Forcing, *Geophys. Res. Lett.*, 2018, 45, 3164-3172

Harper, J.; Humphrey, N.; Pfeffer, T. & Brown, J., *Firn Stratigraphy and Temperature to 10 m Depth in the Percolation Zone of Western Greenland, 2007-2009*, Occasional Paper, Institute of Arctic and Alpine Research, University of Colorado, 2011

Hughes, T. P. & Seligman, G., The temperature, melt water movement and density increase in névé of an Alpine glacier, *Monthly Notices of the Royal Astronomical Society. Geophysical Supplement*, 1939, 4, 616-647

Humphrey, N. F.; Harper, J. T. & Pfeffer, W. T., Thermal tracking of meltwater retention in Greenland's accumulation area, *J. Geophys. Res.*, 2012, 117, F01010

Karlsson, N. B.; Colgan, W. T.; Binder, D.; Machguth, H.; Abermann, J.; Hansen, K. & Pedersen, A., Ice-penetrating radar survey of the subsurface debris field at Camp Century, Greenland, *Cold Reg. Sci. Technol.*, 2019, 165, 102788

MacFerrin, M.; Machguth, H.; van As, D.; Charalampidis, C.; Stevens, C. M.; Heilig, A.; Vandecrux, B.; Langen, P. L.; Mottram, R.; Fettweis, X.; van den Broeke, M. R.; Pfeffer, W. T.; Moussavi, M. S. & Abdalati, W., Rapid expansion of Greenland's low-permeability ice slabs, *Nature*, 2019, 573, 403-407

Machguth, H.; MacFerrin, M.; van As, D.; Box, J. E.; Charalampidis, C.; Colgan, W.; Fausto, R. S.; Meijer, H. A.; Mosley-Thompson, E. & van de Wal, R. S., Greenland meltwater storage in firn limited by near-surface ice formation, *Nature Climate Change*, 2016, 6, 390-393

Miège, C.; Forster, R. R.; Brucker, L.; Koenig, L. S.; Solomon, D. K.; Paden, J. D.; Box, J. E.; Burgess, E. W.; Miller, J. Z.; McNerney, L.; Brautigam, N.; Fausto, R. S. & Gogineni, S., Spatial extent and temporal variability of Greenland firn aquifers detected by ground and airborne radars, *Journal of Geophysical Research*, 2016, 121, 2381-2398

Montgomery, L.; Koenig, L. & Alexander, P., The SUMup dataset: compiled measurements of surface mass balance components over ice sheets and sea ice with analysis over Greenland, *Earth Syst. Sci. Data*, 2018, 10, 1959-1985

Samimi, S.; Marshall, S. J. & MacFerrin, M., Meltwater Penetration Through Temperate Ice Layers in the Percolation Zone at DYE-2, Greenland Ice Sheet, *Geophysical Research Letters*, 2020, 47, e2020GL089211

Vandecrux, B.; MacFerrin, M.; Machguth, H.; Colgan, W. T.; van As, D.; Heilig, A.; Stevens, C. M.; Charalampidis, C.; Fausto, R. S.; Morris, E. M.; Mosley-Thompson, E.; Koenig, L.; Montgomery, L. N.; Miège, C.; Simonsen, S. B.; Ingeman-Nielsen, T. & Box, J. E., Firn data compilation reveals widespread decrease of firn air content in western Greenland, *The Cryosphere*, 2019, 13, 845-859

Reviewer #2 (Remarks to the Author):

Culberg et al \Extreme Melt Season Ice Layers Reduce Firn Per-

meability"

This paper uses airborne radar data to trace an ice layer formed in the Greenland Ice Sheet firn after an extreme melt event in 2012. The existence of this layer is well-known, but our knowledge of the spatial extent and radar reflection

characteristics of the 2012 refrozen-melt layer is greatly extended by the results presented here.

We know that ice layers in a snow cover can change the pattern of water flow within it; field experiments in Greenland have shown reduced "macroscale" hydraulic permeabilities in the presence of ice lenses and layers. However, it is also the case that ice layers need not be totally impermeable barriers; melt-

water can percolate through preferred rapid-flow pathways. This can happen on a relatively small (order 1 m) scale so the authors need to be careful about the interpretation of their remote-sensing data.

The authors explain that the CReSIS accumulation radar produced information about the 2012 ice layer at 15 m intervals along the

flight track. They then define the lateral connectivity of the ice layer by a scoring system that determines whether the layer peak is seen in 3 consecutive radar returns. This allows them to define regions of the ice sheet in which the layer is continuous

on the 15 m scale. This does not necessarily mean that the layer is physically continuous in the sense that there are no vertical pathways through it, but is a reasonable way of defining the geographical extent of the feature.

The authors then make the assumption that the layers are flat and continuous on the 20 m scale in order to estimate the density of the layer seen in each radar return. These densities may be thought of as effective densities on the 20 m scale. It is reasonable to suppose that the "effective density" is to some

extent a measure of the ability of the layer to retard vertical percolation of meltwater; the problem lies in defining what the connection might be, in the absence of ground truth for calibration and validation.

The authors define the "probability of impermeability" as the probability that the observed radar reflectivity is consistent with a layer whose effective density exceeds 0.81 g cm^3 . This value is the approximate bubble close-off density for snow i.e. the density at which an element of snow becomes impermeable.

The implication is that the authors do not allow the possibility of sub-15 m scale density variability and preferred paths through the layer. This assumption really needs much more discussion and justification.

The problem is that, at the heart of the paper, lies the idea that the average of the lateral connectivity and the probability of impermeability can be used to define the percentage reduction in permeability caused by the ice layer. On p.5 the reader is presented with estimates of the average percentage reduction of

permeability caused by the 2012 melt layer for various regions of the ice sheet in terms of means and standard deviations of individual values, without any caveats or estimates of the possible systematic errors arising from the authors' methods and assumptions. This really does seem like a step too far.

On the other hand, the lateral connectivity and estimated layer density can be useful when it comes to comparison of the state of the melt layer in 2013 and 2017. For example, the fact that in south-central Greenland both measures increased over the period can reasonably be interpreted as an indication that the layer has grown in this region.

The data presented in the paper are well-worth having as a contribution to studies of the 2012 extreme melt event and its aftermath and, in the future, if suitable cal/val experiments can be devised, could be interpreted more securely in terms of ice layer permeability. However, as it stands, the paper contains a lot of general discussion of the (undisputed) potential importance of impermeable ice layers attached to statements about the permeability of the 2012 layer which do seem open to question.

Reviewer #3 (Remarks to the Author):

Recent formation of thick ice layers after extreme melt seasons are the topic of the paper – both the existence of these and their continuing thickening and expansion in area. These have a significant

impact on the surface mass balance for the ice sheet by rapidly accelerating the pace of melt run-off in percolation regions. The paper uses some novel analyses of radar data (some of which needs better illustration) and combines it with a widely-used and accepted climate + surface mass balance model, MAR, in interesting and useful ways to describe the process in quantitative terms and track regional differences in the physical characteristics of the expanding ice slab from the 2010 and 2012 extreme melt years.

This paper is worthy of publication, well-written, fairly clear, with interesting graphics and tabular data. I have several minor suggestions that ought to be addressed / considered before acceptance, but in my view the paper should be accepted as valid and useful science.

One minor comment worth noting here is that a lot of the significance of the melt slabs was already published --- this paper should adjust a few sentences just slightly to acknowledge this more. (and for information, I am not a co-author on these earlier studies).

Detailed comments

Abstract

Lines 29-31 – this is a bit of a pet peeve; these are throw-away lines. And in fact there is plenty known and written on these topics. This sentence could be removed with no loss to the paper.

Line 32 – add a horizontal distance – up to 700m higher in elevation, and up to ?50? km inland.
Line 33 – ‘We find that melt layer formation....’ Do you mean ice layer? If you do mean ‘melt layer’ here and in the next sentence, you probably need a sentence that explains what you mean. Melt layer; a layer of saturated snow sitting atop an impermeable ice layer in the upper firn, formed by extensive summer surface melting. It may refreeze in place or flow laterally?

Line 34 – you say ‘precondition’ ... but the melt layers, if I understand correctly, -become- the perched low-permeability horizons, right? Not quite the same as ‘precondition’.

This is a lot like the MacFerrin paper so far. You may need to spend some words differentiating your study from that one, even in the abstract.

Line 49 – add ‘firn’ ---into the porous near-surface snow or firn.’

Line 62 change ‘long-term’ to ‘multi-year’... A long-term (decades or more) excess meltwater production would simply change the facies of the ice sheet to the next one downhill. All these melt slab processes are a manifestation of the very rapid pace of warming and increased melting on Greenland – with a decade climate has reached the point where extreme years build slabs...

Line 68. I think this kind of statement will tend to make people look past the very similar work of Maghuth and MacFerrin.. Your main point is that you’ve extended the kind of analysis they did to a wider area of Greenland, and (below) analyzed the radar profiles in new and interesting ways in conjunction with an SMB model. There are also some very interesting graphical summaries of your point in Figs 3 and 4. But claiming a superlative is just not necessary – the reader does not know where ‘the first’ part of the study ends (it ends at ‘Greenland – wide’).

Line 85 – ‘...quantified the layer lateral connectivity...’. I think this might be better stated as the ‘lateral layer connectivity’... But more importantly, I think this may deserve a figure. To do what you’re saying, you need a more highly resolved radargram than what is shown in the figures you have – a zoom to km -scale or even 100 meter scale? In Line 350-355 you briefly cover how you quantify this – what is the horizontal resolution of the radar profiles? Again, I think a figure might be useful for this.

Line 117 – The current version of MAR is 3.11, quite a ways down the road from this version.

Lines 115- 130 – this seems to be to be the main contribution of the paper – this kind of analysis linking climate (albeit modelled climate) and radar return....

Line130-131 – However, this is overstated. Refreezing is the ‘fundamental mechanism’ in both cases,

of course. You are describing the likely cause of the difference just above – that the deeper firn is quite cold in the north, has a high cold content, and is capable of stopping the melt and refreezing it in a short vertical interval. And ok, I see that you say this in Lines 132-141. This is not fundamentally different, it's different in detail.

Line 142 – I think it is still a faux pas to start a sentence with a number. Could instead start the sentence with 'Extreme but short-lived melt events characterized 2012's...

Line 154-155 – a look at the Greenland Today site at NSIDC (using MAR) shows that the likely year that this happened, for the most part, was 2016. The 2013, 14, and 15 melt seasons were not impressive.

Line 183 – check format for reference 30

Conclusions are good.

Figure 2: you show the firn aquifer extent and ice slab extent in the panels – it would be appropriate to cite Macguth et al., MacFerrin et al., and add in Miller et al., 2020 TCryo.

Figure 3: 3c in particular needs more explanation. 'Standard Deviation of the time series of daily melt -rates- series ... Do you mean the std dev of the year-to-year time series of melt-day totals? Re-write.

Figure 4: is there no space to bring the similar supplemental figure forward?

Ted Scambos

We would like to thank all of the reviewers for their thoughtful and constructive comments which have helped us greatly in improving the manuscript. Please find our detailed responses below.

Reviewer #1:

General remarks

[R1-1] *The conclusions of the manuscript might be valid; however, they seem poorly supported by the data. I agree that Operation IceBridge (OIB) radar measurements show widespread evidence of an ice layer that formed in the firn during the summer 2012. However, I do not fully understand how the study reaches its further conclusions, such as quantifying permeability of the firn, accretion of ice around the 2012 melt layer and identification of “fundamentally different [...] mechanisms” at play (lines 130-131).*

Thank you for these comments. We have worked to clarify both our methods and how we make our interpretations in this revised manuscript. Some general notes on these specific points (these are discussed in more detail in response to the detailed comments):

1) Quantifying permeability of the firn

In our original manuscript, the metric which we named “reduction in permeability” combines the effects of layer density and connectivity such that a continuous solid ice layer would have a maximum score of 1 and unaltered dry firn would have a score of 0. However, multiple reviewers correctly pointed out that this really is a structural metric that tells us something about how the firn structure has changed on average at the scale of the radar footprint and we do not have information we would need to quantitatively link this to a hydraulic property like permeability (see our response to Reviewer #2 for more details). We agree that our choice of name and framing was unfortunate and that truly connecting this metric to permeability would require extensive dedicated field validation studies. To address this, we have done the following:

- Rename this metric “layer prominence” and state for the reader that this is a structural metric that combines the influences of radar-derived density and connectivity
- Discuss the limitations of this metric (e.g., it is spatially averaged, does not account meter-scale variability within the radar footprint, and is not a hydraulic property) and what would be needed to link this metric to hydraulic properties of the ice sheet.
- Explain that this metric provides a quantitative metric of the structure and a qualitative proxy for whether the 2012 melt layer might be able to macroscopically reduce vertical percolation. This is because high density and connectivity will both reduce the number of high-permeability vertical pathways through the firn and is most useful for assessing large spatial trends at the continental scale.

2) Accretion of ice around the 2012 melt layer

In southern Greenland, we reach the conclusion that ice accreted around the 2012 melt layer based on the apparent increase in the radar inferred density of the 2012 melt layer between April 2013 and April 2017. If we have understood the detailed comments in full, it seems that the main point of concern is that our original results suggesting ice accretion in northwest Greenland is not consistent with field observations and that our results in southern Greenland were presented without discussion of the thermal state of the firn or surface melt production in those years. To address this, we discuss both in more detail in response to comments R1-3, R1-9, R1-10, and R1-11 but overall, we agree that our original results were not reasonable in the

northwest and after reprocessing the connectivity data, we believe our results are now consistent with in-situ observations. For the south, we now discuss firn temperatures from observation as well as the modeled melt production between 2013 and 2017 relative to the observed changes in density and place our understanding of our results in the context of field studies on deep/preferential percolation in sub-freezing firn.

3) Identification of different mechanisms as play in ice layer formation

It seems that the main concern on this point is that characterizing the mechanisms as “fundamentally different” is an overreach. What we had hoped to convey in this section of the paper is that it is difficult to find a single climatic variable that can predict where these melt layers form and that the cold content variable shows a good correlation with layer prominence in the northwest and south separately, but not across the whole ice sheet. It seems that given two places with the same firn cold content, one in the NW and one in the south, only one of the locations might also have formed a melt layer. This suggests that there are multiple factors that need to converge to form these layers and the dominant mechanism in each region might be different. For example, in the northwest it might be sharp vertical thermal gradients that prevent deep percolation where in the south it might be high density layers or other ice lenses from previous melt seasons.

In the revised manuscript, we have reworded this sentence to read “Our mapping demonstrates that, rather than heterogeneous densification over the whole firn column as we might expect from consistent deep infiltration¹², refreezing in 2012 consistently formed a laterally coherent melt layer. However, these regional differences in climate and melt layer character suggest that the mechanisms driving this kind of stratification may vary across the ice sheet.”

[R1-2] *These findings might be correct, but they seem poorly supported by the data and they lack critical evaluation. Furthermore, I do not understand why the study relies almost exclusively on remote sensing and modelling: none of the findings seem critically assessed in the context of field measurements. The changing firn of the Greenland ice sheet is currently a very active research field (e.g., Humphrey et al., 2012; Forster et al., 2014; Miège et al., 2016; Graeter et al., 2018; MacFerrin et al., 2019) and a broad variety of field measurements has become available over the last decade (e.g., Harper et al., 2011; Machguth et al., 2016; Montgomery et al., 2018; Vandecrux et al., 2019; Karlsson et al., 2019).*

Thank you for these good points. We discuss extensively in R1-16 and R1-17 why proper in-situ validation of our method is difficult and inherently limited by a lack of field data that is spatially and temporally coincident with the radar collection. However, the point that we should, at a minimum, assess our results in the context of the findings from field measurements is well taken. In this revision, we now discuss our melt layer observations in the context of prior field studies, include Supplementary Figure 2 which shows qualitative comparisons between our results and firn core density profiles, and include firn core density profiles in our interpretation of the layer character differences between northwest and southern Greenland. We have also expanded our discussion of the time series results to discuss how the processes we believe we observe might be occurring in the context of the field observations of firn core stratigraphy, firn temperature, and deep percolation processes. Each of these points is discussed to a greater extent under the corresponding detailed comment below.

[R1-3] *For example, at Camp Century the authors identify a strong reflection from the 2012 melt horizon. Furthermore, it is stated that firn permeability in the area is reduced by 56 ± 14 %. By coincidence, I was present during the drilling of two firn cores (73 and 62 m deep) at Camp Century, in the summer of 2017*

(only a few months after the OIB overflights used in the manuscript). Indeed, the 2012 melt event was clearly recognizable in the core and appears as two ice lenses, each of them about 5 cm thick and about 8 cm apart (Fig. 1, data available upon request at <http://www.campcenturyclimate.dk>). These were the most massive ice lenses observed in the cores (see, e.g., Karlsson et al., 2019) and Culberg et al. identify them successfully from OIB data. However, given that there are only two very thin ice lenses above the 2012 horizon (nearly no melt since 2012; Karlsson et al., 2019) and that firn temperature at the depth of the 2012 lenses is roughly -23 °C (Colgan et al., 2018; also see <http://www.campcenturyclimate.dk/ccd/data.html>; data available at <http://promice.org/>), I doubt that the connectivity of the ice lenses changed after 2012.

Thank you for this example and for the firn core photos from Camp Century – we now include a comparison of our radar-inferred density and layer depths with the firn density profile from the B73 core at Camp Century which shows very nice agreement (Supplementary Figure 2). This point concerning the firn temperatures is important and motivated us both to include field observations in our discussion as well as re-assess the methods we used to estimate layer connectivity. We found that the radargram flattening we applied prior to estimating connectivity tended to artificially reduce the layer connectivity. This particularly affected the 2013 connectivity estimates as the surface was generally more heterogenous/rough and the proximity of the melt layer to the surface made automatic surface tracking less reliable. We have reprocessed all of the data (2017 and 2013) without this flattening procedure and updated our results accordingly. After this reprocessing, we find only a small change in connectivity on the northwest transect between 2013 and 2017 which is likely not significant, particularly given that we see no change in layer density. This brings our results into a consistent state with the field observations. We have edited line 240 in the manuscript to explain that we assess that there were no changes to the 2012 melt layer along the northwest transect and discuss why this is consistent with the field observations from Colgan, et al (2018) and Karlsson, et al (2019).

[R1-4] *Lines 36-38 read “Our results demonstrate that a single extreme melt season can reshape the near surface hydrology of Greenland’s interior, altering the ice sheet’s long-term response to surface melting.” (lines 36-38, emphasis added). In its current form, I do not think the study provides sufficient evidence to support this statement. My criticism (i) touches upon the use or quantification of “permeability” (see detailed comments) and (ii) the relevance of isolated (in the vertical domain) ice lenses in a nearly melt-free environment. In the current climate, these ice lenses get buried by accumulating snow, their potential to block meltwater percolation seems to be of theoretical nature (as illustrated on the example of Camp Century where five years after the 2012 melt event the lenses are at 5 m depth and there is no evidence of any other major melt event after 2012).*

The criticism of our use of permeability is a valid one and we have significantly revised how we present that metric to make it clear that this is not a hydraulic parameter and the constraints on its interpretation. See R1-1 for a more in-depth discussion of our changes.

We agree that relevance of vertically isolated melt lenses in low melt environments is also worthy of more discussion than it received in our original manuscript. We have expanded our discussion to clarify the following points:

- Our mapping of the 2012 melt layers only tells us how much the structure of the firn has been altered at this horizon as a result of refreezing and does not, by itself, imply hydrologically significant impacts.
- The time series of changes in density and connectivity can, however, provide evidence for how the 2012 melt layer has interacted with subsequent surface melting and therefore whether or not it is of sufficiently low permeability to impede future flow.

- We discuss that, as you have pointed out here, in cold, in low melt areas like Camp Century, the melt layer is vertically isolated from the surface hydrology and therefore has little to no impact on future refreezing.
- This stands in contrast with the southwest and south-central Greenland where we do see changes in both layer density and connectivity that suggest more melt has refrozen at this horizon. In this specific region, the ability of these layers to reduce vertical percolation pathways through the firn and encourage further ice aggregation seems not to be theoretical and we now discuss this possibility in the context of observed firn temperatures and percolation depths.
- We make sure to point out throughout the rest of our discussion that the ultimate impact of extreme melt season ice layers depends on the regional SMB regime and that it is the combination of a melt layer and a favorable melt to accumulation ratio that make ongoing growth and continued hydrological interactions possible.

Ultimately, we have revised the abstract to read: “These melt layers reduce vertical percolation pathways, and, under appropriate firn temperature and surface melt conditions, encourage further ice aggregation. Therefore, the frequency of extreme melt seasons relative to the rate at which pore space and cold content regenerates above the most recent melt layer may be a key determinant of the firn’s multi-year response to surface melt.”

Detailed Comments

[R1-5] Lines 78 – 79: *“Density contrast of 0.3 g cm⁻³”. I do not know whether this value is also valid for the Camp Century location, but the measurements at Camp Century roughly confirm such a contrast: firn density in the cores was measured at 0.1 m vertical resolution, the three 0.1 m section that cover the 2012 ice lenses in the 73 m core have densities of 0.709, 0.508 and 0.64 g cm⁻³. Density of the surrounding firn is roughly 0.45 g cm⁻³.*

This density contrast would apply at Camp Century and it is gratifying to see that there is agreement here. We now include a comparison of our radar-inferred density and layer depths with the firn density profile from the B73 core at Camp Century which shows very nice agreement (Supplementary Figure 2).

[R1-6] Lines 87 – 90: *Unfortunately, I do not understand how “permeability” is calculated (also the description in the methods section is unclear to me). Please also consider that there is rich evidence that ice lenses are basically never “impermeable” (e.g. Hughes and Seligman, 1939, pages 635-640; Samimi et al., 2020). Any extended ice lens seems to have sufficient discontinuity (cf. Brown et al., 2011) to allow for percolation of water. Machguth et al. (2016) even show evidence of minor amount of meltwater percolating to >5 m depth in an ice slab environment.*

Please see our response to R1-1 with regards to the permeability metric. Additionally, we have updated the methods sections to better explain how we calculate the metric we now refer to as “layer prominence” (line 539). The point about percolation through temperate ice and locally discontinuous ice layers is well taken and we no longer refer the melt layer as impermeable in the manuscript.

[R1-7] Lines 111 – 113: *“This substantial reduction in permeability suggests that the melt layer has the potential to impact near-surface hydrologic processes over a wide swath of the ice sheet where the layer exists.” See my major comments and the comment above. In my opinion, such a statement needs more evidence or at least an in-depth discussion.*

We no longer make this statement at this point in the manuscript as we agree that it is not well supported by mapping data alone. Instead, we make the point that our data shows that refreezing following the 2012 melt season substantially altered the structure of the firn across large parts of the ice sheet. This qualitatively implies that where sufficient future surface melting occurs and is capable of percolating to depth, these structural changes can alter the macroscopic vertical flow pathways for that melt. We reserve further discussion of the hydrological implications of melt layer formation for the discussion of the time series data which better supports the idea that some regions of the ice sheet have seen ongoing interactions between the surface hydrology and 2012 melt layer.

[R1-8] Lines 132-141: *I agree to these findings. However, I would not label them “fundamentally different mechanisms” (line 131). I expect formation of a less distinct ice layer where meltwater percolates through firn that has seen more regular percolation in the past.*

We have reworded this sentence to read “Our mapping demonstrates that, rather than heterogeneous densification over the whole firn column as we might expect from consistent deep infiltration, refreezing in 2012 consistently formed a laterally coherent melt layer. However, these regional differences in climate and melt layer character suggest that the mechanisms driving this kind of stratification may vary across the ice sheet.”

[R1-9] Lines 152-156: *See my major remarks. I understand that the Camp Century example cannot simply be considered representative for all of the ice layer formation in the north-west (although the location is highlighted in the manuscript). Nevertheless, more information is needed to explain how this would work.*

As discussed in R1-3, we agree that deep percolation seems implausible at Camp Century and is not supported by the field data. After reprocessing our data, we believe that our results are now in agreement with this assessment, and we discuss all of these points in the context of the field measurements in the manuscript (line 236).

[R1-10] Lines 156-161: *It is stated earlier in the manuscript that in 2017 the 2012 ice layer was 3 to 12 m below the surface, depending on accumulation rates. I assume that 12 m is more valid for the south where accumulation is higher? Please explain whether the suggested growth in ice thickness at substantial depth (up to 6 m on average) is possible. It might be possible, but needs to be discussed in the context of meltwater percolation depth under given amount and frequency of melt after 2012 as well as observed firn temperatures.*

We now discuss our observations in southwest and south-central Greenland in the context of modeled melt production between 2013 and 2017, firn core stratigraphy, thermistor/thermocouple observations of firn temperatures where available, and other field observations of percolation depths (lines 243-270). Unfortunately, current observations of percolation depths and the associated temperatures are from regions much lower in the percolation zone than our ice layer observations. However, in light of those studies, transient breakthrough along preferential pathways in the top 2 m seems most likely for this region, suggesting that melt layer modification in the 2013 and 2014 melt seasons is at least plausible. Additionally, 2016 was a somewhat high melt year (2-3 standard deviations above 1980-2011 mean) which might have allowed for more extensive percolation on the western side of the transects where the melt layer was only ~3.5m below the surface in that year and where we see the largest density changes.

[R1-11] Lines 165-171: *See my major comments and the comment above: This important statement needs more evidence.*

We agree with your overarching point that the existence of an ice layers does not inherently imply hydrological significance. However, we do believe we have shown that under the right conditions, there are places in southern Greenland where the 2012 melt layer continued to interact with the surface hydrology in subsequent years and where the 2012 melt layer itself formed in relation to the pre-existing 2010 ice layer. We have clarified that this is not a blanket statement for all ice layers everywhere in the ice sheet, but pertains to the subset where climate conditions allow for sufficient percolation, and does not necessarily require a truly impermeable melt layer. This discussion now reads: “The multi-year persistency and continued growth of these refrozen features highlights a feedback mechanism where, under appropriate firn temperature and surface melt conditions, existing melt layers can promote the ongoing development of low-permeability horizons by reducing vertical percolation pathways and concentrating refreezing near the surface, even if the original melt layer is not itself fully impermeable. This is analogous to the ice lens aggregation mechanism proposed for the growth of ice slabs, but in these interior regions, an extreme melt season can be the key catalyst that initiates this aggregation, rather than multi-year excess melt.”

[R1-12] Lines 171-173: *I basically agree to this statement. However, mentioning the extraordinary melt season of 2019 might be confusing in the context of this study. All firn changes described refer to the melt seasons of 2016 or earlier.*

We mention the 2019 melt season to make the point that the 2012 melt season was not an isolated event and that these kinds of extreme seasons seem to be re-occurring somewhat frequently. We have added an additional discussion sentence at this point in the manuscript to clarify this point and to note that this is interesting precisely because of your point about vertically isolated ice layers. While the 2012 ice layers might not remain relevant forever as it gets buried under new accumulation, if extreme seasons continue to occur and form new melt layers, the balance between rates of melt, accumulation, and frequency of layer formation becomes a potentially important determinant of the firn capacity.

[R1-13] Lines 195-196: *I agree that bursts of melt play an important role in ice layer formation. However, the study also states that firn cold-content is relevant (here defined as the total cold content over the top 20 m, lines 388-390). The cold content of the top 20 m is strongly controlled by climate.*

Agreed. What we hoped to convey here is that there is not a single long-term climate predictor of where melt layers will form in extreme years. For example, regions in southern Greenland with the same firn cold content as in NW Greenland saw minimal layer formation because 2012 melt production was much lower. So, while the climate sets very important conditions that allow for formation, the specifics of melt volume and variability in the extreme season are also very important. We softened this sentence to read “...melt layer formation may be strongly governed by local, short timescale thermal and hydrologic processes *in addition* to the mean climate state.”

[R1-14] Lines 196-197: *This statement is not fully clear to me: “At the same time, our findings reveal that extreme melt layers are an ice-sheet-wide phenomenon that interacts with the entire firn hydrologic system.”*

What we hoped to point out here is that while there are these very regional differences in the drivers of formation, we still see that this ice layer has formed in nearly every sector of the ice sheet, across multiple facies, and with interactions with both ice slabs and firn aquifers. We have clarified this

sentence to read “At the same time, our findings reveal that extreme melt layers may form in all regions of the Greenland Ice Sheet, with interactions spanning facies from the dry snow zone to the wet snow zone “.

[R1-15] Figure 3: *The subplots lack labelling a), b) and c). How are the different regions of Greenland defined?*

We have corrected the subplot labeling and added dashed boxes to Fig 2 which show the regions which were defined as northwest and southern Greenland in this analysis.

[R1-16] Methods, data calibration and layer detection: *I might be wrong, but would more recent in situ firn measurements be helpful to calibrate and validate interpretation of the OIB data?*

Yes, absolutely! However, this is difficult to do in practice because while there are a large amount of in-situ firn measurements available, to use them for good calibration and validation of radar data, we need, at minimum, for those observations to spatially and temporally coincident with the radar flights in regions where we expect to see the signatures we are trying to validate. For this study, this means we need observations from after October 2012, collected within 10 meters of a radar flight line and within a few months of the overflight (certainly before the start of the next melt season) in a region where we expect to see this melt layer. Of the publicly available data that we’re aware of, there are exactly two firn cores at one location that meet all of those criteria. Even if we are very generous with the flight line overlap criterion (allowing cores to be multiple kilometers away) since we expect this to be a fairly continuous signature, there are still only three cores, all clustered around the southern saddle, plus the Camp Century core. It would be fantastic to see a funded project to collect these kinds of field measurements in a dedicated campaign (perhaps similar to the altimeter validation transects). We could learn a great deal more about how the small-scale variability within the firn is expressed in the radar data, the possibility of detecting subtler refreezing signals, and the connections between radar signatures and the hydraulic properties of the firn with proper field calibration and validation experiments.

We have a made a point in this manuscript revision to point out where new fieldwork would improve and build upon our methods and how the remote sensing and in-situ measurements are fundamentally complementary. We hope this paper is in fact a motivation to fund the future field studies needed for robust in-situ validation of these kinds of methods.

[R1-17] Lines 346, 357, 367: *Please explain why data interpretation seems to be validated solely against idealized model output. In the end, the OIB data are used to make statements on real-world processes on the Greenland ice sheet. And there is a large amount of in situ firn measurements available.*

As we described in comment R1-16 above, the lack of spatially and temporally coincident observations makes it very difficult for us to develop robust methods that rely entirely on comparisons to in-situ measurements.

1) Layer Detection

To determine if refrozen ice layers produce a distinct signature in Accumulation Radar data and if our algorithm can detect them, we would need to correlate known ice layers with radar data. With only a single firn core available to us, any interpretation simply cannot be robust. We have no way of knowing whether how the radar system would respond to ice layers

of different thickness, densities, depths, or spacing than those represented in this core or how the response we see may be confounded by the background density or other factors at this particular site. Additionally, since the radar response depends on the average properties of the whole radar footprint, that single core may not even be totally representative of the bulk properties that the radar is responding to. A generalizable interpretation of the radar data strictly by comparison to in-situ measurements would ideally require measurements across the radar footprints at tens to hundreds of different sites.

For all these reasons, we think that electromagnetic modeling of the radar's response to ice layers in the firm is a more robust way to determine the system's sensitivity to these layers. This allows us to test layers of different thickness, spacing, and density against different background densities to understand under what conditions we would expect ice layers to be detectable in the Accumulation Radar data.

To keep the methods concise, we did not discuss the details of this modeling since it has already been published (Culberg & Schroeder, 2020a; Culberg & Schroeder, 2020b). However, it is worth explaining that this modeling is fundamentally founded on in-situ observations. We showed in Culberg & Schroeder, 2020a that we can do an excellent job of simulating Accumulation Radar traces using in-situ density profiles as the input to our electromagnetic model. Unfortunately, we need density measurements at the ~5mm scale to do this effectively, limiting us the North Greenland Traverse cores which were analyzed with GAP or DEP DECOMP methods at AWI. We used the high-resolution density profiles from six of those cores to build a statistical density model so we can generate density profiles with realistic annual layering. We set the mean density profile deterministically using information from lower resolution cores closer to our areas of interest. For our simulations, we then run hundreds of thousands of trials using these density profiles with ice layers inserted into the density profile. We use a density of refrozen ice determined by Machguth, et al (2016) from their in-situ measurements. This all allows us to gather robust statistics on the performance of our layer detection algorithm.

While we agree that no model is perfect and our is no exception, it is worth noting that our methods are highly likely to result in more false negatives, but few to no new false positives when transferred to real radar data. This is because we rely on the relative brightness of radar returns from ice layers to detect them. There are a myriad of factors that might make ice layers less bright in real radar data (layer roughness, discontinuities, variable density, attenuation, receiver noise, etc) all of which would make them less likely to be detected. There are really no real-world conditions which would make the layers appear brighter than in the idealized model and thus more likely to be falsely detected.

2) Reflectivity Inversion

In order to invert the radar reflectivity for layer density, we essentially need a look-up table that relates the two variables. We also know from electromagnetic theory that this will be inherently coupled with layer thickness and background firm density, making this an underdetermined inverse problem. This makes it particularly difficult to establish this inverse relationship from in-situ data comparisons alone. Unless we're lucky enough that our observations span this entire parameter space (which seem nearly impossible), we won't have robust knowledge of how good our solution is. Using the models described above, we can build this look-up table by running the full parameter space, so that we know that whole range of solutions that are consistent with the radar data we observe. But this is one of the reasons we choose to report the minimum consistent density, as it makes the point that even evaluated against our most conservative metric, this melt layer is quite significant.

So, all of that being said, what we can and should do is compare our detection and inversion results to in-situ measurements afterwards to see if we produce results that are at least broadly consistent with the field observations. Some caution must be exercised in making these comparisons, as the radar measurements average firn properties over a 20m horizontal footprint but are very sensitive to the vertical configuration, compared with firn core density measurements which sample a centimeter-wide area but averages densities in the vertical.

To this end, we have made the following changes to the revised manuscripts

- Supplementary Figure 4 now includes firn core density profiles from each of these regions which confirm our interpretation of the radar layer character.
- We have added Supplementary Figure 2 which compares firn core density profiles with our results from the nearest radar trace. Ideally this comparison would include detailed core stratigraphy logs, but those are not (at least publicly) available for these cores to the best of our knowledge.
- We discuss the reasons for our use of simulations more thoroughly in the methods section (lines 485-495, 526-531, 544-556).

[R1-18] Extended Data Figure 2: *In general, the handling of geographical locations is confusing. Looking at this figure, I am surprised that a radar transects slightly north of 70 °N is labelled "southwest Greenland". But I agree with the naming if it were somewhere stated or shown how the authors subdivided Greenland for the purpose of this study.*

Thank you for pointing this out. In Figure 2, we have added outlines that show the regions which we categorize as northwest and southern Greenland for the climatological analysis shown in Figure 3. In Extended Data Figure 2 (now Supplementary Figure 3 in the revised document), we now show a different transect which is more properly from the southwest as typically defined so that we can compare the radar line to nearby firn core.

[R1-19] Extended Data Figure 3: *Similar to above, for the most part this transect seems to be located in the east, not in the west.*

We have moved this figure into Figure 4, but we have adjusted the figure legend and text to refer to the transects in Figure 4 by its label (transect B) in general and only use the descriptor of "southwest" when discussing the specific portion of the transect where we see increased density.

Reviewer #2:

[R2-1] *This paper uses airborne radar data to trace an ice layer formed in the Greenland Ice Sheet firn after an extreme melt event in 2012. The existence of this layer is well-known, but our knowledge of the spatial extent and radar reflection characteristics of the 2012 refrozen-melt layer is greatly extended by the results presented here.*

Thank you.

[R2-2] *We know that ice layers in a snow cover can change the pattern of water flow within it; field experiments in Greenland have shown reduced "macroscale" hydraulic permeabilities in the presence of ice lenses and layers. However, it is also the case that ice layers need not be totally impermeable*

barriers; meltwater can percolate through preferred rapid-flow pathways. This can happen on a relatively small (order 1 m) scale so the authors need to be careful about the interpretation of their remote-sensing data.

The authors explain that the CReSIS accumulation radar produced information about the 2012 ice layer at 15 m intervals along the flight track. They then define the lateral connectivity of the ice layer by a scoring system that determines whether the layer peak is seen in 3 consecutive radar returns. This allows them to define regions of the ice sheet in which the layer is continuous on the 15 m scale. This does not necessarily mean that the layer is physically continuous in the sense that there are no vertical pathways through it, but is a reasonable way of defining the geographical extent of the feature. The authors then make the assumption that the layers are flat and continuous on the 20 m scale in order to estimate the density of the layer seen in each radar return. These densities may be thought of as effective densities on the 20 m scale. It is reasonable to suppose that the "effective density" is to some extent a measure of the ability of the layer to retard vertical percolation of meltwater; the problem lies in defining what the connection might be, in the absence of ground truth for calibration and validation.

The authors define the "probability of impermeability" as the probability that the observed radar reflectivity is consistent with a layer whose effective density exceeds 0.81 g cm³. This value is the approximate bubble close-off density for snow i.e. the density at which an element of snow becomes impermeable. The implication is that the authors do not allow the possibility of sub-15 m scale density variability and preferred paths through the layer. This assumption really needs much more discussion and justification.

This is an excellent summary of what is possible to interpret from our data and how it should be framed for the reader. In the manuscript, we now explicitly state these assumptions for the reader at line 108 clarifying that the radar data only allows us to derive spatially averaged metrics over the ~20m radar footprint which does not preclude variability or layer discontinuities at smaller scale. We have also reshaped our interpretation and discussion of the "reduction in permeability" metric so that it is better aligned with these realities, changes which we discuss more extensively below.

[R2-3] *The problem is that, at the heart of the paper, lies the idea that the average of the lateral connectivity and the probability of impermeability can be used to define the percentage reduction in permeability caused by the ice layer. On p.5 the reader is presented with estimates of the average percentage reduction of permeability caused by the 2012 melt layer for various regions of the ice sheet in terms of means and standard deviations of individual values, without any caveats or estimates of the possible systematic errors arising from the authors' methods and assumptions. This really does seem like a step too far.*

We were a bit overenthusiastic about the possible scientific implications of what we were seeing, and naming our combined density and continuity metric "reduction in permeability" was an unfortunate choice. We fundamentally agree that establishing a quantitative relationship between our radar-derived metrics and the hydraulic properties of the ice sheet would require dedicated field validation studies to look at the sub-meter variability in density and microstructure over the radar footprint and how the bulk average of those properties is related to in-situ observed permeability in the context of preferential flow paths and firn thermal state. We have made the following revisions in the manuscript which we believe present a more appropriate interpretation of our results:

- 1) We retain the combined metric where we average the lateral layer continuity score and the probability that the layer density exceeds pore close-off. However, we rename the metric "layer prominence". We think that this metric is still useful in assessing the continental-scale spatial

trends in formation and gives a sense for where the firn structure has undergone the most significant alteration from refreezing in 2012. However, we agree that it should not be presented as hydraulic property.

- 2) Before presenting our quantification of layer prominence, we explain that this metric should not be interpreted as a quantification of the hydraulic properties of the ice sheet but is rather a structural metric that represents spatially averaged changes in firn structure as a result of refreezing (line 146).
- 3) When discussing the potential hydrological implications of these structural changes, we explain that our metric is a strictly qualitative proxy for regions where the melt layer could present a macroscopic (> 20m scale) impediment to vertical percolation since permeability is inversely related to density and high lateral connectivity will reduce the number of high-permeability pathways for vertical flow (line 152).
- 4) Where we discuss the large-scale spatial trends in layer prominence (line 159), we soften our language to clarify that layer prominence helps us identify areas where the 2012 refreezing has most significantly altered the firn structure, which has the potential to alter pathways for future melt – although the precise nature of that interaction is not something we can quantify from this data alone.

[R2-4] *On the other hand, the lateral connectivity and estimated layer density can be useful when it comes to comparison of the state of the melt layer in 2013 and 2017. For example, the fact that in south-central Greenland both measures increased over the period can reasonably be interpreted as an indication that the layer has grown in this region.*

Thank you. We agree that the time series data is more useful in assessing whether (and where) this melt layer presents an impediment to vertical flow. We have added a sentence at the beginning of this section to make that connection explicit. We think that your comments also highlight a useful point – in fact, we likely do not need a perfectly impermeable foundation for these ice layers to continue to grow. Like the ice lens aggregation mechanism that produces ice slabs, slowing or partially reducing percolation pathways can be enough to see continued ice aggregation. Extreme melt seasons just have the potential to kick start this kind of process in regions of the ice sheet that otherwise wouldn't see enough refreezing in their general climate for aggregation to be a serious concern. We have added this discussion point to the manuscript.

[R2-5] *The data presented in the paper are well-worth having as a contribution to studies of the 2012 extreme melt event and its aftermath and, in the future, if suitable cal/val experiments can be devised, could be interpreted more securely in terms of ice layer permeability. However, as it stands, the paper contains a lot of general discussion of the (undisputed) potential importance of impermeable ice layers attached to statements about the permeability of the 2012 layer which do seem open to question.*

Thank you. We absolutely agree, and would love to see those kinds of field studies funded. We believe that our revised manuscript now makes it clear that we believe these extreme melt season ice layers can have important hydrological implications because of this time-series evidence for continued growth in the south and ice slab connections in the northwest, not because we can claim to map their hydraulic properties in the first half of the paper.

Reviewer #3:

[R3-1] *Recent formation of thick ice layers after extreme melt seasons are the topic of the paper – both the existence of these and their continuing thickening and expansion in area. These have a significant impact on the surface mass balance for the ice sheet by rapidly accelerating the pace of melt run-off in percolation regions. The paper uses some novel analyses of radar data (some of which needs better illustration) and combines it with a widely-used and accepted climate + surface mass balance model, MAR, in interesting and useful ways to describe the process in quantitative terms and track regional differences in the physical characteristics of the expanding ice slab from the 2010 and 2012 extreme melt years.*

This paper is worthy of publication, well-written, fairly clear, with interesting graphics and tabular data. I have several minor suggestions that ought to be addressed / considered before acceptance, but in my view the paper should be accepted as valid and useful science.

Thank you.

[R3-2] *One minor comment worth noting here is that a lot of the significance of the melt slabs was already published --- this paper should adjust a few sentences just slightly to acknowledge this more. (and for information, I am not a co-author on these earlier studies).*

Thank you for pointing this out. We have implemented the detailed suggestions towards this point you offer below and have edited our abstract and introduction to make it clearer how our study differs from or expands upon Machguth and MacFerrin's work on ice slabs. We also point out in the discussion of our time series results the way in which the multi-year growth of these ice layers is analogous to the mechanism already proposed for ice slabs, but with a different catalyst (single season extreme melt) in these relatively lower melt regions.

Detailed comments

Abstract

[R3-3] *Lines 29-31 – this is a bit of a pet peeve; these are throw-away lines. And in fact there is plenty known and written on these topics. This sentence could be removed with no loss to the paper.*

We dropped this sentence from the abstract.

[R3-4] *Line 32 – add a horizontal distance – up to 700m higher in elevation, and up to ?50? km inland.*

Added.

[R3-5] *Line 33 – ‘We find that melt layer formation....’ Do you mean ice layer? If you do mean ‘melt layer’ here and in the next sentence, you probably need a sentence that explains what you mean. Melt layer; a layer of saturated snow sitting atop an impermeable ice layer in the upper firn, formed by extensive summer surface melting. It may refreeze in place or flow laterally?*

This is a useful point of clarification. We used “melt layer” rather than “ice layer” because, particularly in the south, this 2012 refreezing may not be a singular ice layer but a complex conglomeration of many

ice layers (see Supplementary Figure 4, for example). We use melt layer to refer to the entire complex of refrozen ice formed in 2012. At line 84 we now clarify what our subsequent use of the term “melt layer” is intended to mean.

[R3-6] *Line 34 – you say ‘precondition’ ... but the melt layers, if I understand correctly, -become- the perched low-permeability horizons, right? Not quite the same as ‘precondition’.*

In order to shorten the abstract to the required length we ended up dropping this sentence entirely.

[R3-7] *This is a lot like the MacFerrin paper so far. You may need to spend some words differentiating your study from that one, even in the abstract.*

We have added some discussion at the end of the introduction clarifying that the rapid formation of these ice layers in relatively lower melt regions suggests that their formation conditions, character, and multi-year evolution are likely to be significantly different from what we understand about ice slabs, making these question interesting in their own right.

[R3-8] *Line 49 – add ‘firn’ --...into the porous near-surface snow or firn.’*

Fixed in the text.

[R3-9] *Line 62 change ‘long-term’ to ‘multi-year’... A long-term (decades or more) excess meltwater production would simply change the facies of the ice sheet to the next one downhill. All these melt slab processes are a manifestation of the very rapid pace of warming and increased melting on Greenland – with a decade climate has reached the point where extreme years build slabs...*

Fixed in text.

[R3-10] *Line 68. I think this kind of statement will tend to make people look past the very similar work of Maghuth and MacFerrin. Your main point is that you’ve extended the kind of analysis they did to a wider area of Greenland, and (below) analyzed the radar profiles in new and interesting ways in conjunction with an SMB model. There are also some very interesting graphical summaries of your point in Figs 3 and 4. But claiming a superlative is just not necessary – the reader does not know where ‘the first’ part of the study ends (it ends at ‘Greenland – wide’).*

We dropped “... the first...”

[R3-11] *Line 85 – ‘...quantified the layer lateral connectivity...’. I think this might be better stated as the ‘lateral layer connectivity’... But more importantly, I think this may deserve a figure. To do what you’re saying, you need a more highly resolved radargram than what is shown in the figures you have – a zoom to km -scale or even 100 meter scale? In Line 350-355 you briefly cover how you quantify this – what is the horizontal resolution of the radar profiles? Again, I think a figure might be useful for this.*

We adjusted this wording in the text. The horizontal resolution of the radar data is approximately 20 meters and we have added additional details at line 108 describing this resolution and explaining to the reader that our radar derived metrics are inherently spatial averages over this length scale and cannot be interpreted as smaller scales. We have also added Supplementary Figure 1 which shows a close-up of the layer structure in the radargrams and kernel we use to quantify the connectivity.

[R3-12] Line 117 – *The current version of MAR is 3.11, quite a ways down the road from this version.*

Indeed. We used a MARv3.5.2 run because it is, to our knowledge, the most recent publicly available data set that includes the full 3D firn data that we need to calculate the integrated firn cold content and it seemed more appropriate to keep the comparisons consistent within one model run rather than mix and match across different versions.

[R3-13] Lines 115- 130 – *this seems to be to be the main contribution of the paper – this kind of analysis linking climate (albeit modelled climate) and radar return....*

Thank you for pointing this out.

[R3-14] Line 130-131 – *However, this is overstated. Refreezing is the ‘fundamental mechanism’ in both cases, of course. You are describing the likely cause of the difference just above – that the deeper firn is quite cold in the north, has a high cold content, and is capable of stopping the melt and refreezing it in a short vertical interval. And ok, I see that you say this in Lines 132-141. This is not fundamentally different, it’s different in detail.*

We dropped the “fundamental” in this sentence and rewrote it to read “Our mapping demonstrates that, rather than heterogeneous densification over the whole firn column as we might expect from consistent deep infiltration¹², refreezing in 2012 consistently formed a laterally coherent melt layer. However, these regional differences in climate and melt layer character suggest that the mechanisms driving this kind of stratification may vary across the ice sheet.”

[R3-15] Line 142 – *I think it is still a faux pas to start a sentence with a number. Could instead start the sentence with ‘Extreme but short-lived melt events characterized 2012’s....*

Edited to read “The 2012 melt season ...”.

[R3-16] Line 154-155 – *a look at the Greenland Today site at NSIDC (using MAR) shows that the likely year that this happened, for the most part, was 2016. The 2013, 14, and 15 melt seasons were not impressive.*

This is an interesting point, as Reviewer #1 also raised the point that by 2016, this ice layer was buried 4-7 m below the surface, which would require a fair bit of deep percolation for surface melt to reach it. The observed change in density would, without considering compaction densification, require between 0.5-35% of the total 2013-2016 modeled melt to reach and refreeze at this layer, or nearly all of the 2013 melt. However, the layer was nearer to the surface in these earlier years, making it easier for water to reach given the relatively cold firn temperatures. So from this time series alone, it’s very difficult to say whether we are looking at an integrated effect over several light melt years or a large dump of water just from 2016. But we now discuss these points in the manuscript at lines 243-270 where we evaluated our observed changes in light of the modeled climate and field observations of percolation depths.

[R3-17] Line 183 – *check format for reference 30*

This is a peculiarity of the referencing format where it avoids applying superscript reference numbers to other numbers – hence the parenthetical citation after the number 2010.

[R3-18] *Conclusions are good.*

Thank you.

[R3-19] *Figure 2: you show the firn aquifer extent and ice slab extent in the panels – it would be appropriate to cite Macguth et al., MacFerrin et al., and add in Miller et al., 2020 TCryo.*

Good catch! We have added citations to the caption – MacFerrin, et al for the ice slabs and Miede, et al for the firn aquifers since the datasets we plot are direct outputs/supplements to those papers and Abdalati for the passive microwave melt boundaries.

[R3-20] *Figure 3: 3c in particular needs more explanation. ‘Standard Deviation of the time series of daily melt -rates- series ... Do you mean the std dev of the year-to-year time series of melt-day totals? Rewrite.*

We have clarified this to read “Correlation with the standard deviation of the modeled daily surface melt production from January 1, 2012 to December 1, 2012.” We have also added a parenthetical pointer to the Methods where we have space to discuss this a bit more extensively. But the gist is that MAR is run at daily time steps, so for each grid cell have daily melt production in mmWe for each day of the year in 2012. We take the standard deviation of this time series of melt production at each location.

[R3-21] *Figure 4: is there no space to bring the similar supplemental figure forward?*

We have reworked the figure layout to bring forward the supplemental figure into the main figure 4.

REVIEWERS' COMMENTS

Reviewer #1 (Remarks to the Author):

Review Culberg et al., revised manuscript submitted to Nature Communications

General remarks

Culberg et al. present a strongly modified and revised manuscript. In my opinion, the changes have clearly improved the study. The revised manuscript has become a very valuable contribution to an important topic of research. I believe that the revised study now provides a clear and thorough estimate of the potential importance of extreme melt events in shaping Greenland's firn cover. The research raises a number of important questions and has the potential to become an important inspiration to future research.

I have only a small number of minor remarks.

Detailed Comments

Lines 27: Elsewhere the authors write of "low-permeability". Maybe use a similar term here too? Totally impermeable is extremely unlikely as any natural ice layer, even ice slabs, maintain a low level of permeability (due to discontinuities, in the case of ice slabs probably also due to cracks).

Lines 213-214: This appears extreme (I probably misunderstood something and would be thankful for clarifications). This means that, e.g., at the elevations where MAAT is -10°C , there was five to ten times more melt in the northwest? Is this in absolute numbers or relative to average annual melt at the locations? I am also surprised because elsewhere it is stated that the latent heat /cold content ratio (for a given layer prominence) is only about half in the northwest compared to the south. If, however, absolute melt (I assume equal to latent heat release) was up to 10 times larger in the northwest, then cold content in the north must be up to 20 times larger than in the south? And because cold content scales linearly with temperature (assuming density and thickness of the firn pack considered are equal) then this is difficult to achieve.

Line 262: Not fully clear what "that temperature condition" means.

Line 314: I do not understand what is meant by "These down-glacier interactions".

Supplementary Figure 2: Thank you very much for including so much field data. There might be minor issue with the citation of the data. I understand the authors obtained all the data shown either from the Camp Century project or from the SumUp database. Some of the data shown are indeed from ref. 8, however, the remaining data have been first published in ref. 9 (I think also the FirnCover project belongs to ref. 9). I suggest citing refs. 8, 9 and 20 (and of course ref. 16 for Camp Century data).

Supplementary Figure 2: Maybe specify more precisely which cores are shown, especially there are several EKT cores, I assume it is the 2013 one?

Reviewer #2 (Remarks to the Author):

This revised version of the paper has been greatly improved and the authors have dealt thoughtfully with my criticisms of the first version. They have now made it clear that their "layer prominence" metric is to be considered as a qualitative measure of a possible reduction in firn permeability. Furthermore they have added material linking their remotely-sensed data to field measurements and set their work in the context of previous work in the same field. The result is an interesting paper with novel results which will add to our understanding of the effects of a warming climate on the Greenland Ice Sheet.

Reviewer #3 (Remarks to the Author):

The authors have addressed the comments extensively and rather thoroughly within the bounds of

what can be determined from airborne data and limited ground truth. I am not as pessimistic as some of the other reviewers about the overall idea that an extensive thick ice layer at high elevation will modify the densification of the upper firn in the subsequent melt years, and potentially the transition to run-off with further extreme melt events. The 2019 melt event, mentioned briefly, offers an opportunity for further testing of the idea.

I continue to think that the paper deserves publication in Nature Communications, and will likely be cited as the literature regarding evolving firn condition in a rapidly-warming environment builds.

I skimmed the revised paper, noted a few small items:

Line 31 – keep the words ‘ice layer’. There’s no doubt that an ice layer formed as a result of the intense melt – it’s there, in both radar and field data. It just may not be as continuous as you inferred in the earlier version.

Line 41 – suggest you add ‘...aggregation at the same ice layer depth.’ Or something like that.

Line 43 – suggest you add ‘..multi-year response to an intense episode of surface melt.’

Line 134 suggest ‘lateral layer continuity’

Thanks again to all of the reviewers for the thoughtful comments that helped us greatly in improving this manuscript.

Reviewer #1 (Remarks to the Author):

Review Culberg et al., revised manuscript submitted to Nature Communications

General remarks

[R1-1] *Culberg et al. present a strongly modified and revised manuscript. In my opinion, the changes have clearly improved the study. The revised manuscript has become a very valuable contribution to an important topic of research. I believe that the revised study now provides a clear and thorough estimate of the potential importance of extreme melt events in shaping Greenland's firn cover. The research raises a number of important questions and has the potential to become an important inspiration to future research.*

I have only a small number of minor remarks.

Thank you! Your detailed and thoughtful comments on our original submission were instrumental in putting this manuscript on firm scientific footing.

Detailed Comments

[R1-2] *Lines 27: Elsewhere the authors write of "low-permeability". Maybe use a similar term here too? Totally impermeable is extremely unlikely as any natural ice layer, even ice slabs, maintain a low level of permeability (due to discontinuities, in the case of ice slabs probably also due to cracks).*

This is a good point and we have updated the text accordingly.

[R1-3] *Lines 213-214: This appears extreme (I probably misunderstood something and would be thankful for clarifications). This means that, e.g., at the elevations where MAAT is -10 °C, there was five to ten times more melt in the northwest? Is this in absolute numbers or relative to average annual melt at the locations? I am also surprised because elsewhere it is stated that the latent heat /cold content ratio (for a given layer prominence) is only about half in the northwest compared to the south. If, however, absolute melt (I assume equal to latent heat release) was up to 10 times larger in the northwest, then cold content in the north must be up to 20 times larger than in the south? And because cold content scales linearly with temperature (assuming density and thickness of the firn pack considered are equal) then this is difficult to achieve.*

Thanks, this is a good point of clarification. Part of the discrepancy is that since the MAT in many northwest regions where this ice layer appears it quite a bit colder than where it appears in the south, this particular discussion implicitly compares areas in the northwest where a melt layer formed to areas in the south without a distinguishable melt layer. Because those are relatively low melt regions in the south (low enough for no distinguishable layer to form) the apparent difference in melt is quite high. This difference in melt between the northwest and south also decreases quite a bit as the firn warms; at a MAAT of -18°C the model shows pretty much the same amount of melt in both regions.

The latent heat/cold content ratio only looks at areas where melt layers actually formed, so it is looking at a somewhat different subset of regions on the ice sheet. Additionally, because we are comparing regions with the same layer prominence between the northwest and south, we are, for example, implicitly comparing regions with a MAT of -23°C to those with a MAT of -19°C (and

accompanying difference in surface melt driven simply by elevation rather than regional differences). As a result, drawing a direct line between these two comparisons is tough. We've updated the text at line 179 to explicitly associate the difference in melt production with the appropriate MAT range and clarify that we're trying to explain why a melt layer was able to form in this relatively cold region in the northwest, but not in regions of similar climate in the southwest.

[R1-4] *Line 262: Not fully clear what "that temperature condition" means.*

We were trying to refer back to the possibility of percolation through $-10\text{ }^{\circ}\text{C}$ firn as observed in other locations, as we use the depth of the $-10\text{ }^{\circ}\text{C}$ isotherm as an indication of reasonable percolation depths. We've clarified this sentence to read "Given these observations, firn temperature measurements from the early 2000s at the NASA-SE GCNet site^{34,35} and the 2016 GreenTRACS traverse^{15,36} suggest that meltwater percolation to depths of up to 2 meters during the melt season could be possible on these transects."

[R1-5] *Line 314: I do not understand what is meant by "These down-glacier interactions".*

We have clarified this sentence to read: "These down-glacier connections to ice slabs and firn aquifers highlight how the multi-year impact of extreme melt season ice layers depends on the local SMB regime, with outcomes ranging from vertical isolation to the development of near-surface ice slabs."

[R-6] *Supplementary Figure 2: Thank you very much for including so much field data. There might be a minor issue with the citation of the data. I understand the authors obtained all the data shown either from the Camp Century project or from the SumUp database. Some of the data shown are indeed from ref. 8, however, the remaining data have been first published in ref. 9 (I think also the FirnCover project belongs to ref. 9). I suggest citing refs. 8, 9 and 20 (and of course ref. 16 for Camp Century data).*

Thanks for catching this. The appropriate citation for the FirnCover data was a bit unclear, since the latest SUMup release still cites an "in prep" paper for a lot of the later cores. Your suggestion to include ref. 9 is a good one and we have updated the citations accordingly.

[R1-7] *Supplementary Figure 2: Maybe specify more precisely which cores are shown, especially there are several EKT cores, I assume it is the 2013 one?*

Was this comment intended to reference the EKT core shown in Supplementary Figure 4 (rather than Sup. Fig. 2 where we already have the cores labeled by date and location)? Assuming this is the case, we have updated the figure caption to give the date that the EKT core was collected.

Reviewer #2 (Remarks to the Author):

[R2-1] *This revised version of the paper has been greatly improved and the authors have dealt thoughtfully with my criticisms of the first version. They have now made it clear that their "layer prominence" metric is to be considered as a qualitative measure of a possible reduction in firn permeability. Furthermore they have added material linking their remotely-sensed data to field measurements and set their work in the context of previous work in the same field. The result is an interesting paper with novel results which will add to our understanding of the effects of a warming*

climate on the Greenland Ice Sheet.

Thank you! We greatly appreciated your insightful comments on our original submission.

Reviewer #3 (Remarks to the Author):

[R3-1] *The authors have addressed the comments extensively and rather thoroughly within the bounds of what can be determined from airborne data and limited ground truth. I am not as pessimistic as some of the other reviewers about the overall idea that an extensive thick ice layer at high elevation will modify the densification of the upper firn in the subsequent melt years, and potentially the transition to run-off with further extreme melt events. The 2019 melt event, mentioned briefly, offers an opportunity for further testing of the idea.*

I continue to think that the paper deserves publication in Nature Communications, and will likely be cited as the literature regarding evolving firn condition in a rapidly-warming environment builds.

Thank you! We really appreciated your constructive comments and overall enthusiasm for our work.

I skimmed the revised paper, noted a few small items:

[R3-2] *Line 31 – keep the words ‘ice layer’. There’s no doubt that an ice layer formed as a result of the intense melt – it’s there, in both radar and field data. It just may not be as continuous as you inferred in the earlier version.*

We considered changing this wording but given that we use melt layer throughout the rest of the abstract and paper when referring to this feature, we felt it was better to maintain consistency in our wording here.

[R3-3] *Line 41 – suggest you add ‘...aggregation at the same ice layer depth.’ Or something like that.*

Good suggestion, updated in text.

[R3-4] *Line 43 – suggest you add ‘..multi-year response to an intense episode of surface melt.’*

In this case, we intentionally wanted to refer to surface melting as a whole, as part of what we hope to convey in this paper is that the firn structural changes in extreme melt years alter the future response of the ice sheet to surface melting that occurs even in typical melt seasons.

[R3-5] *Line 134 suggest ‘lateral layer continuity’*

Updated in text to ‘lateral layer connectivity’ to be consistent with the phrasing throughout the rest of the text.